



# Multivariate statistical modelling of extreme coastal water levels and the effect of climate variability: a case study in the Netherlands

Victor M. Santos[1,*], Mercè Casas-Prat[2,*], Benjamin Poschlod[3,*], Elisa Ragno[4], Bart van den Hurk[5], Zengchao Hao[6], Tímea Kalmár[7], Lianhua Zhu[8], and Husain Najafi[9]

[1]Department of Civil, Environmental and Construction Engineering, and National Center for Integrated Coastal Research, University of Central Florida, Orlando, Florida, USA
[2]Climate Research Division, Science and Technology Directorate, Environment and Climate Change Canada, Toronto, Ontario, Canada
[3]Department of Geography, Ludwig-Maximilians-University, Munich, Germany
[4]Faculty of Civil Engineering and Geosciences, Delft University of Technology, Delft, The Netherlands
[5]Deltares, Delft, The Netherlands
[6]College of Water Science, Beijing Normal University, Beijing, China
[7]Department of Meteorology, Faculty of Science, Institute of Geography and Earth Sciences, Eötvös Loránd University, Budapest, Hungary
[8]Key Laboratory of Meteorological Disaster, Ministry of Education; Joint International Research Laboratory of Climate and Environment Change, Nanjing University of Information Science & Technology, Nanjing, China
[9]Department of Computational Hydrosystems, Helmoltz Centre for Environmental Research-UFZ, Leipzig, Germany
[*]These authors contributed equally to this work.

**Correspondence:** V.M. Santos (vmalagon@Knights.ucf.edu)

**Abstract.** The co-occurrence of (not necessarily extreme) precipitation and surge can lead to extreme inland water levels in coastal areas. In a previous work the positive dependence between the two meteorological drivers was demonstrated in a case study in the Netherlands by empirically investigating an 800-year time series of water levels, which were simulated via a physical-based hydrological model driven by a regional climate model large ensemble. In this study, we present and

test a multivariate statistical framework to replicate the demonstrated dependence and the resulting return periods of inland water levels. We use the same 800-year data series to develop an impact function, which is able to empirically describe the relationship between high inland water levels (the impact) and its driving variables (precipitation and surge). In our study area, this relationship is complex because of the high degree of human management affecting the dynamics of the water level. By event sampling and conditioning the drivers, an impact function was created that can reproduce the water levels maintaining

an unbiased performance at the full range of simulated water levels. The dependence structure between the driving variables is modeled using two- and three-dimensional copulas. These are used to generate paired synthetic precipitation and surge events, transformed into inland water levels via the impact function. The compounding effects of surge and precipitation and the return water level estimates fairly well reproduce the earlier results from the empirical analysis of the same regional climate model ensemble. The proposed framework is therefore able to produce robust estimates of compound extreme water levels for a highly

managed hydrological system.

In addition, we present a unique assessment of the uncertainty when using only 50 years of data (what is typically available from observations). Training the impact function with short records leads to a general underestimation of the return levels as





water level extremes are not well sampled. Also, the marginal distributions of the 50-year time series of the surge show high
variability. Moreover, compounding effects tend to be underestimated when using 50 year slices to estimate the dependence
pattern between predictors. Overall, the internal variability of the climate system is identified as a major source of uncertainty
in the multivariate statistical model.

## 1 Introduction

Floods, wildfires, and heatwaves typically result from the combination of several physical processes (e.g., Baldwin et al.,
2019; Manning et al., 2019). Such processes are not necessarily extreme or hazardous when occurring in isolation, but they
can lead to significant impacts when occurring altogether, or in a narrow time range (Pörtner et al., 2019; Zscheischler et al.,
2018). Extreme events resulting from the combinations of physical drivers are referred to as compound events, and can be
classified into different (not entirely exclusive) categories (Zscheischler et al., 2020). These compound climate extremes are
receiving increasing attention because of their disproportionate economic, societal, and environmental impacts, and because
traditional univariate approaches can lead to strongly biased estimates of the associated risks (Zscheischler and Seneviratne,
2017). However, many challenges still lay ahead in order to properly understand, and predict, the complex chain of drivers
that leads to compound events. Estimating the dependencies among drivers is challenging mainly due to the limited amount of
data available, especially for rare events (Zscheischler et al., 2018). Moreover, the definition of multivariate extremes is not as
straightforward as in the univariate case. A paradigm shift from a classical top-down approach adopted in many climate studies
towards an impact-centric perspective is needed (Zscheischler et al., 2018).

This study is motivated by a near flooding event in 2012 in Lauwersmeer in the Netherlands that clearly can be classified
as a compound event (van den Hurk et al., 2015). This multivariate event was characterized by a high inland water level ex-
ceeding predefined warning levels and resulted from the joint occurrence of heavy precipitation on an already wet soil and
a high storm surge impeding gravitational drainage over several consecutive tidal periods. In terms of the categorization of
Zscheischler et al. (2020), this event can be classified as multivariate, pre-conditioned and temporally compounding, which
illustrates the complexity of this near flooding event. van den Hurk et al. (2015) empirically assessed the return periods as-
sociated to compound extreme water levels with a single model initial-condition large ensemble (SMILE) of regional climate
model (RCM) simulations covering 800 years under present-day climate conditions. They demonstrated a positive dependence
between storm surge and heavy precipitation and showed that the probability of occurrence of these extreme water levels can
be greatly underestimated if such dependence is omitted.

SMILEs are a physically based approach to increase the size of the database and therefore increase the number of simulated
extreme compound events. Apart from van den Hurk et al. (2015), SMILEs have been applied as tool to investigate compound
events by e.g. Zhou and Liu (2018), Khanal et al. (2019a) and Poschlod et al. (2020). With the aim to obtain methods computa-



tionally less expensive than numerical simulations, statistical models have been used to model compound events and estimate
their probability of occurrence. In some specific cases, bi- or multi-variate distributions can be derived directly from physical
properties (e.g. the joint distribution between wave height and wave periods in wind-sea states as a function of wave steepness
(de Waal and van Gelder, 2005)). However, these are often limited to idealized or very specific settings and rely heavily on the
selection of the marginal distributions. In contrast, copula-based methods (Sklar, 1959) have the advantage to capture the de-
pendence between a set of variables independently from their marginal distributions (Genest and Favre, 2007), which explains
why they have become a widely used approach nowadays.

Compound flooding in coastal settings often originates from a combination of storm-driven waves and surges, and blocked
discharge of terrestrial water from e.g. intense precipitation or snow melt. Meteorological conditions can lead to a (nearly)
simultaneous occurrence of storm surge or waves and a discharge peak when the area that generates the discharge is located
close to the coast. These types of events occur in many coastal regions across the globe (Ward et al., 2018; Couasnon et al.,
2019) and their associated impacts strongly depend on the catchment features and the characteristics of the storms (Wahl et al.,
2015). For discharge peaks originating from remote precipitation or snow melt inputs (for instance in larger river systems)
delays between the surge and discharge peaks are usually due to the finite travel speed of the discharge wave (Khanal et al.,
2019b; Klerk et al., 2015). In recent years, several copula-based studies have been carried out to study compound flooding
events in coastal areas at different spatial scales (e.g. Couasnon et al., 2018; Moftakhari et al., 2019; Jane et al., 2020). For
example, Bevacqua et al. (2015) developed and implemented a conceptual statistical model to quantify the risk of compound
floods that result from the combination of storm surge and high river runoff in Ravenna (Italy). At regional scale, Wahl et al.
(2015) assessed the historical changes in the compound flooding due to precipitation and storm surge in US cities and identified
a significant increase in the number of compound events over the past century in major coastal cities. Accounting for climate
change projections, Bevacqua et al. (2019) showed how global warming can increase the probability of compound coastal
flooding in Northern Europe. At a global scale, Couasnon et al. (2019) provided a perspective of the compound flood potential
from riverine and coastal flood drivers, which highlighted the complexity and large regional variability of such dependence
structures. Dependence between ocean wave heights and storm surges was recently investigated by Marcos et al. (2019) at
global scale, showing that 55% of the world coastlines face compound storm surge wave extremes.

This study explores whether a copula-based model can reproduce the findings in van den Hurk et al. (2015) for the Lauw-
ersmeer reservoir, using the same 800-year climate dataset as reference. Two novel aspects are addressed in our analysis. First,
we investigate the strong impact of the definition and selection of the predictors based on the meteorological drivers and their
interaction on the resulting water levels. An extra complication is generated by the strong human management of the water
system. This type of flooding event has been explored rarely in the literature (most flooding studies cover natural systems),
despite the growing relevance of flood risk management in many low-lying managed areas (Pörtner et al., 2019) where sea
level rise increases flood frequency (Moftakhari et al., 2017; Taherkhani et al., 2020). Second, we explore for the first time (to
our knowledge) the effect of internal climate variability on copula-based compound event analysis. We investigate the effect of
using a 50-year subset of data on the estimation of dependence structures (and other elements involved in the compound event
analysis), ultimately assessing the accuracy of the estimation of return levels. This is particularly relevant as most compound



climate extreme studies are based on observations or simulated time-slices with lengths well under 50 years (e.g. Ganguli and
Merz, 2019; Wahl et al., 2015; Zheng et al., 2013). The global study of Ward et al. (2018) showed that most available datasets
of overlapping discharge-surge have a median duration of 36 years, with shorter to no observed records in most of Africa,
South America and Asia.

## 2   Data and study area

The study area comprises the water management unit Noorderzijlvest (1440 km$^2$) situated in the north of the Netherlands,
which has an average altitude close to mean sea level height. The Lauwersmeer reservoir stores excessive water before it drains
into the North Sea by gravity during low tides. In January 2012, a combination of heavy and prolonged rainfall on saturated soil
during high sea level conditions (blocking the free drainage) led to extreme inland water levels accompanied by precautionary
implications such as evacuation. Both precipitation and storm surge associated to this event were mild extremes (with return
periods of about 10 years, respectively), but the inland water reached unusually extreme levels.
95         In terms of the underlying meteorological patterns, extreme winds with long fetch leading to high surges typically occur in
October-December as a result of deep and extensive low-pressure systems moving from the North Atlantic region to central or
Northern Scandinavia (van den Hurk et al., 2015). Most extreme precipitation events occur during the summer months linked
to slow-moving medium-sized low-pressure systems over northern Germany or southern Denmark (van den Hurk et al., 2015).
High water levels are caused by the interaction between these two patterns, which mostly occur in July-October. Addition-
ally, Ridder et al. (2018) found that the majority of these types of compound events are accompanied by the presence of an
atmospheric river over the Netherlands.
        van den Hurk et al. (2015) empirically estimated the return periods of inland water level by applying a physically based
modelling chain. They used the climate simulations of the 16-member ensemble of the RCM KNMI RACMO2 (van Meijgaard
et al., 2008; Van Meijgaard et al., 2012) driven by the global climate model (GCM) EC-EARTH 2.3 (Hazeleger et al., 2012).
Forced by historical emissions, the GCM was run from 1850 to 2000 with 16 different perturbations of initial atmospheric
conditions. This ensemble was dynamically downscaled by the RCM at 12 km horizontal resolution for transient runs from
1951 to 2000, resulting in 800 years of historic climate. After bias-adjustment, these regional simulations were then used
to drive RTC-Tools, a hydrological management simulator (Schwanenberg et al., 2015) generating the corresponding inland
water level time series at hourly resolution. As the 16 50-year simulations only differ by the initial atmospheric conditions of
the driving GCM, the variability of the 16 time series can be interpreted as model representations of the internal variability of
the climate system (Deser et al., 2012; Hawkins and Sutton, 2009).
        To assess compounding effects, van den Hurk et al. (2015) constructed a randomized ensemble of independent drivers by
shuffling the time series of model generated precipitation and storm surge in a way that preserved climatological characteristics
but removed the correlation between surge and precipitation. After adding the tidal cycle, the corresponding water levels were
derived by forcing RTC-Tools with these shuffled time series of precipitation and total surge. van den Hurk et al. (2015)
concluded that the return period associated to the extreme 2012 water level was almost three times larger for shuffled data than





for the original data, which indicated the presence of a compounding effect of precipitation and surge on water level (which was also shown by comparing the empirical joint probability density functions of the original and shuffled time series). However, the dependence of surge and precipitation was weaker for the largest water level events, which were dominated by specific

neap tide conditions with a low tidal range and consequently high values of the low tides.

## 3 Methods

### 3.1 Conceptual model

The statistical model for estimating inland water level has been developed following four consecutive steps:

1. Characterization of the compound event with a predictand, representing the so-called "impact" (water level), and a set of
predictors (conditioned to the impact variable) representing the underlying drivers (precipitation and surge) of extreme water levels.

2. Development of an impact function that relates the predictand and predictors defined in step (1).

3. Modelling of the joint probability distribution of the predictors, which implies finding the probability distributions to model their marginal behavior, and identifying the best copula(s) to model their dependence structure.

4. Estimating the return water levels by randomly generating a large number of paired precipitation and storm surge synthetic events from the joint distribution obtained in step (3), which is converted to annual maximum water levels with the impact function fitted in step (2).

  To reproduce the findings of van den Hurk et al. (2015), including the effect of the dependence between precipitation and surge on return levels, this procedure is applied to both the original dataset and the shuffled data (see Section 2). More details

of each step are provided in the remainder of this Section. The design of the analyses has followed an iterative process, with repeated feedbacks between the different steps. The selection of the predictors plays a crucial role in the consecutive steps and the performance of the statistical modeling framework. Specifically, the performance of the impact function is highly sensitive to this selection and has been a strong driver for the final choice of predictors.

### 3.2 Selection of predictands and predictors

The series of annual maxima of inland water level ($WL_{\max}$) is chosen as predictand to represent the impact and used to reproduce the return plots of van den Hurk et al. (2015). In the process of predictors selection, three aspects were taken into consideration: (1) the underlying physically driving processes, including the proper representation of the compound nature of surge and precipitation; (2) the human management practices controlling the inland water level dynamics in RTC-tools (Section 2); (3) the memory of the physical system, including lags in the occurrence of drivers that might potentially affect the

magnitude of the impact.







**Figure 1.** Composite of flooding drivers and associated water level response for the 2D (a) and 3D (b) cases, computed using all 800 annual maxima events. Solid lines represent the median of all values at a given time, whereas the shaded areas depict the values between the $5^{th}$ and $95^{th}$ percentiles. Vertical lines indicate the time windows used for the selected predictors (see Table 1).

To illustrate the rationale behind the selection of the predictors, the composite of all 800 $WL_{\max}$ and the underlying drivers is visualized in Fig. 1. Peaks in precipitation and total storm surge are preceding the occurrence of the annual $WL_{\max}$. Opening and closing the gates of the reservoir leads to periodic fluctuations of the inland water level. The gates are opened during the low tide to lower the inland water level. If the ocean water level exceeds the inland water level, the gates stay closed and the inland water level rises due to collection of water from the surrounding watershed. For most of the 800 annual maximum events, the gates stay closed for several subsequent tidal cycles (see Fig. 1).

We explored statistical models of two and three dimensions to account for multiple predictors. For the 2D case, we choose the following predictors: the accumulated precipitation over 12 days prior to $WL_{\max}$, noted as $P_{12d,\mathrm{acum}}$, and the minimum


**Table 1.** Selected predictors for the 2D and 3D cases. Note that total surge is the sum of surge plus tide

| 2D case | 3D case |
|---|---|
| $P_{12d,\mathrm{acum}}$: accumulated precipitation over 12 days prior to $WL_{\max}$ | $P_{12d,\mathrm{acum}}$: accumulated precipitation over 12 days prior to $WL_{\max}$ |
| $S^T_{36h,\min}$: minimum total surge over 36 h prior to $WL_{\max}$ | $S_{72h,\mathrm{mean}}$: mean surge over 72 h prior to $WL_{\max}$ |
| | $T_{12h,\min}$: minimum tide over 12h prior to $WL_{\max}$ |

total surge over the 36 h prior to $WL_{\max}$, noted as $S^T_{36h,\min}$. For the 3D case, the precipitation predictor is the same as

in 2D case, but the total surge is separated into the astronomical tide and the non-tidal residual (hereinafter tide and surge, respectively). With this separation we investigate whether the difference in controlling physical processes of tide and surge affects the depiction of the dependency structure causing compounding effects. In particular, we consider the mean surge over 72 h prior to $WL_{\max}$, noted as $S_{72h,\mathrm{mean}}$, and the minimum tide over 12h prior to $WL_{\max}$, noted as $T_{12h,\min}$ (see Table 1). The time periods of aggregation, as well as the choice of applying the arithmetic mean, minimum or the sum, were iteratively

optimized according to the performance of the impact function and its reproduction of the return period curves (see Section 3.3 and 3.4). The composite plots (Fig. 1) guided this iteration process.

The iterative process of predictor selection led to interesting insights about the physical processes behind these compound events. In terms of precipitation, Fig. 1 shows that the duration of the median peak of accumulated precipitation prior to $WL_{\max}$ is about 5 days, which agrees with the relevant temporal range of precipitation directly affecting the inland water

level identified by van den Hurk et al. (2015). Instantaneous contribution of precipitation to inland water levels due to direct rainfall on the reservoir surface is small and therefore a time lag is needed to capture the contributions from surface runoff, streamflow and interflow caused by rainfall over the whole catchment. However, the impact function performs better for a longer aggregation time period (12 days). We argue that the precipitation prior to 5 days helps to better capture the system memory induced by soil moisture storage, as early rainfall can affect $WL_{\max}$ by saturating the soil. Indeed, one of the factors

contributing to the largest event in 2012 was soil saturation caused by above normal rain in the preceding weeks (van den Hurk et al., 2015). This is shown by the $95^{th}$ percentile precipitation envelope in Fig. 1 that has a peak lasting more than 5 days and has a non-zero plateau for a time lag above 9–10 days.

For the 3D case, the level of the low tide during the antecedent 12-hourly cycle to $WL_{\max}$ is clearly identified as a potential predictor. It varies over time due to astronomical cycles and thus contributes to the timing of the reservoir drainage. The

contribution from the surge is better captured by taking the average over the previous 72h, which perfectly matches the duration of the surge peak observed in Fig. 1b (for both mean and extreme percentiles). When the total surge is considered as one single variable (2D case), a trade-off between the contribution of surge and tide is achieved by considering the minimum total surge over an intermediate time period of 36 h. Figure 1a shows that for most of the 800 events the reservoir gates were closed for at least three tidal cycles (equaling 36 h).

Due to our impact-focused approach (see Section 3.1), the chosen predictors are conditioned to $WL_{\max}$. This deviates from other studies in which an $n$-way sampling approach is followed (i.e. conditioning to one of the (extreme) driving variables at a time) (e.g. Ward et al., 2018). This procedure is usually followed when information about the impact variable is limited





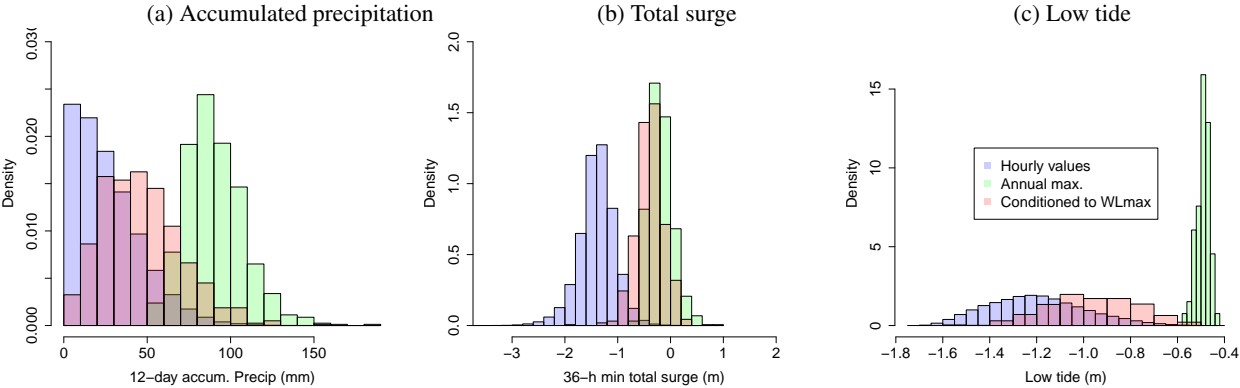

**Figure 2.** Density histograms for precipitation (a), total surge (b), and low tide (c) associated to all hourly time series (blue), to selected predictors (conditioned to $WL_{\max}$) (pink), and to the corresponding univariate annual maxima (green).

and/or when the focus is on identifying the driver that contributes most to compounding effects. Conditioning the drivers on the impact variable guarantees an optimal training of the impact function (Section 3.3) and all extreme water level events

are captured, including those that might not result from the combination of extreme univariate events. Figure 2 compares the distributions of $P_{12d,\mathrm{acum}}$, $S^T_{36h,\min}$ and $T_{12h,\min}$ to the distribution of the corresponding univariate annual maxima. The selected predictors have notably lower values than the corresponding annual maxima, especially for precipitation and tide variables. The corresponding surge events are closer to their annual maxima, which agrees with the dominant role of this water level driver, as seen in Section 3.3.

## 3.3 Impact function

The impact function is designed to reproduce $WL_{\max}$ given a set of predictors (see Section 3.2). We explored different approaches, including multiple linear regression (MLR), random forests (RF) (Meinshausen, 2006) and artificial neural networks with stochastic gradient descent for regression (NN) (He et al., 2015; Phan, 2015). The different regression models are evaluated by means of the root-mean-square error (RMSE), the mean absolute error (MAE), the linear (Pearson's) correlation

coefficient $r$ and the error associated to return level estimates. This procedure was carried out iteratively for different sets of predictors in order to minimize the deviations between the $WL_{\max}$ simulated by the RTC-Tools and the $WL_{\max}$ estimated via the impact functions.

For the 2D case (Table 1), all impact function approaches simulate inland water levels with an RMSE of 9 cm or less, an MAE of 7 cm or less and $r$ greater than 0.7 (see Fig. S2 in the Supplementary Material (SM)). RF exhibits the best performance

by means of $r$=0.88, MAE=4 cm and RMSE= 6 cm. However, none of these approaches reproduce the extreme water levels exceeding 0 m, which have the largest impact (see Fig. S4 in SM) due to the optimization process of the regression models, which uses a cost function penalizing the squared error of the estimated water level for each of the 800 annual maxima. The 800 annual maxima are not evenly distributed across the range of water levels between -0.5 m and 0.22 m. 82 % of the samples





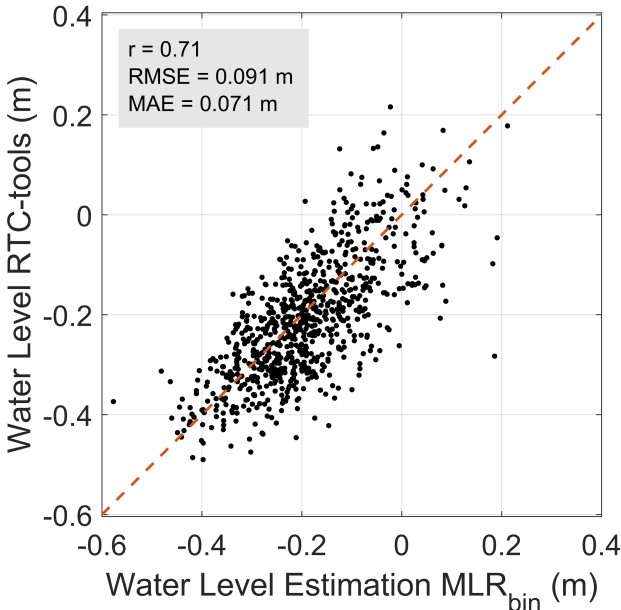

**Figure 3.** $WL_{\max}$ obtained by RTC-Tools vs. $WL_{\max}$ obtained using MLR with sampling approach for the 2D case (see Table 1).

feature water levels below -0.1 m and 94 % of the events show water levels below 0 m. Hence, the optimized regression models

are biased to reproducing $WL_{\max}$ between -0.5 m and -0.1 m.

To overcome the underestimation of the most extreme events, we apply a bin-sampling strategy to train the impact function, iteratively optimizing the number of bins and samples per bin. For the regression models based on machine-learning (RF, NN), this bin-sampling does not increase the performance, as a simple combination of the bootstrapped parameters is not straightforward. Consequently, we opt for MLR as the model of choice to define the impact function. All 800 values are

divided into 12 classes ("bins") according to their inland water level and distributed in 5 cm steps (see Table 2). From each of these bins, 10 samples (9 for the highest bin) are randomly drawn and the parameters of the MLR are optimized for the subset. To avoid any bias due to the randomized selection, this procedure is bootstrapped 1000 times and the mean of the resulting parameters is taken for the final impact function. For MLR a combination of the linear regression factors of the 1000 random runs can well be constructed by applying the arithmetic mean. This results in the final two-dimensional linear regression:

$$WL_{\max} = -0.1639 + 0.3998 \cdot S^T_{36h,\min} + 0.0027 \cdot P_{12d,\mathrm{acum}} \qquad (1)$$

The comparison of $WL_{\max}$ simulated by the RTC-Tools and $WL_{\max}$ estimated via Eq. 1 is shown in Fig. 3. After standardization of the predictors by $\widetilde{X} = (X - \overline{X})/X^{\mathrm{sd}}$, where $\overline{X}$ and $X^{\mathrm{sd}}$ are the corresponding mean and standard deviation, the dominant role of surge compared to precipitation is evident:





**Table 2.** Distribution of the bin-sampling classes.

| bin | WL1 | WL2 | WL3 | WL4 | WL5 | WL6 | WL7 | WL8 | WL9 | WL10 | WL11 | WL12 |
|---|---|---|---|---|---|---|---|---|---|---|---|---|
| WL (m) | <-0.4 | (-0.4,-0.35) | (-0.35,-0.3) | (-0.3,-0.25) | (-0.25,-0.2) | (-0.2,-0.15) | (-0.15,-0.1) | -(0.1,-0.05) | (-0.05,0) | (0 0.05) | (0.05,0.1) | >0.1 |
| # samples | 31 | 55 | 109 | 122 | 136 | 123 | 82 | 63 | 32 | 27 | 11 | 9 |

$$WL_{\max} = -0.1932 + 0.1033 \cdot \widetilde{S}^T_{36h,\min} + 0.0639 \cdot \widetilde{P}_{12d,\text{acum}} \tag{2}$$

For the 3D case (Table 1), we obtained:

$$WL_{\max} = -0.2645 + 0.4652 \cdot S_{72h,\text{mean}} + 0.3434 \cdot T_{12h,\min} + 0.0028 \cdot P_{12d,\text{acum}} \tag{3}$$

This expression has a slightly larger $r$, and lower RMSE and MAE (see Fig. S3). However, the performance of the return level estimations is slightly worse (generally more tendency to underestimate) for the 3D case (Fig. S4 vs. Fig. S5).

### 3.4 Joint probability density function and return levels

The joint distribution of the selected predictors is modelled via a copula function (Sklar, 1959; Nelsen, 2007) (see Section 1 of SM). The selection of the marginal distributions and the dependence structure of the predictors is crucial for a robust assessment of extreme inland water levels. The overall methodology to obtain the return plots is similar between the 2D and 3D cases (see Section 3.1) and implemented as follows. 1) To separate marginal and dependence analysis, data are ranked and transformed to uniform in the unit (hyper)-square using rank statistics; 2) copula family and parameters are fitted to 230 these uniform data with the maximum pseudo-likelihood estimator (Kojadinovic and Yan, 2010); 3) a total of 40 copula types are considered (VineCopula R package, version 2.3.0) selecting the one leading to the lowest Akaike information criterion (AIC) (Schepsmeier et al., 2015). The adequacy of the selected copula model is assessed using a goodness-of-fit test based on Kendall's processes (Genest et al., 2009; Wang and Wells, 2000); 4) Suitable marginal distributions for the (unranked) defined predictors are identified, testing a wide range of distributions commonly used in hydrologic analysis and selecting the one with 235 the best fit (lowest AIC; Sakamoto et al., 1986); 5) the joint probability distribution of the considered predictors is obtained with the best fitted copula(s) and marginals; 6) simulated events from this joint distribution are obtained by sampling uniform data from the copulas and converting to real units with the previously fitted marginals; 7) Finally, the obtained synthetic samples are used to estimate inland water levels via the impact function explained in Section 3.3. Note that the fitted marginals are intentionally not used for the copula fitting in order to make the choice of the copula(s) totally independent from the choice of 240 the marginal(s) (Genest and Favre, 2007).

Once water levels have been calculated, the associated return periods are obtained using Weibull plotting positions (Makkonen, 2006). Compounding effects are assessed by comparing the return value/period curve obtained by fitting the copula model and the marginals to the dependent and the shuffled (independent) data (Section 2). In our analysis, copula models generate





many synthetic events of paired precipitation and surge (up to 100.000) to produce stable return level estimates of inland water

level up to a 10.000-year return period. Although producing a 10.000-year data set from 800 years of empirical data entails

dealing with large uncertainties, especially for the highest return levels, we chose that number because it establishes the standard level of protection in many places in the Netherlands, especially those exposed to severe flooding (Bouwer and Vellinga, 2007).

## 4   Results and discussion

The results of the statistical modelling framework are presented here. We find that the model with three predictors (3D case), i.e., precipitation, surge, and tide, does not generally outperform the model with two predictors (2D case), i.e., precipitation and total surge, (see Table 1). Therefore, results of the 2D case are presented here, leaving most of results of the 3D case in the SM.

### 4.1   Dependence structure between S and P

In order to better understand the underlying factors leading to $WL_{\max}$, this Section explores the dependence structure between surge and precipitation for the 2D case using Kendall's $\tau$ correlation (Kendall, 1938) and the joint PDF (probability density function) of $S^T_{36h,\min}$ and $P_{12d,\text{acum}}$. Different sources of variability are assessed, with a special focus on the internal variability of the climate system.

#### 4.1.1   Interpretation of $\tau$: dependence vs. independence

Since we are interested in those combinations of precipitation and surge that, together, lead to high water level, the Kendall's rank correlation $\tau$ between $S^T_{36h,\min}$ and $P_{12d,\text{acum}}$ is investigated. For the dependent data set, it amounts to -0.05, differing from zero correlation at the 95% significance level. To further investigate the compound nature of the two predictors, the same correlation is calculated using the shuffled (independent) data. In this case, $\tau$ amounts to -0.15.

The negative $\tau$ between $S^T_{36h,\min}$ and $P_{12d,\text{acum}}$ is arguably related to the positive contribution of both surge and precipitation

to $WL_{\max}$ and therefore the negative slope of the $WL_{\max}$ isolines as a function of these predictors: lower values of one driver can be compensated by higher values of the other driver to generate a given water level. This is illustrated with a simple theoretical example in Section 2 of SM. The rank correlation of the dependent case exceeds $\tau$ of the independent case by +0.10, which arguably indicates a positive dependence pattern between surge and precipitation. Fig. 4 shows the joint PDF obtained by our statistical model (see Section 3.4). Similarly, the shaded orange area highlights the increased probability of

having both extreme $S^T_{36h,\min}$ and $P_{12d,\text{acum}}$ (leading to extreme water levels) as obtained from the original data, in comparison to the independent case. This agrees with the findings of van den Hurk et al. (2015) obtained empirically.

In summary, as a result of the conditioning on WL, the correlation between the defined predictors (the explanatory variables of the impact function) does not duplicate the dependence between drivers (precipitation and surge) leading to extreme water levels. Such conditioning complicates the interpretation of the dependence structure and compound effects, but opti-





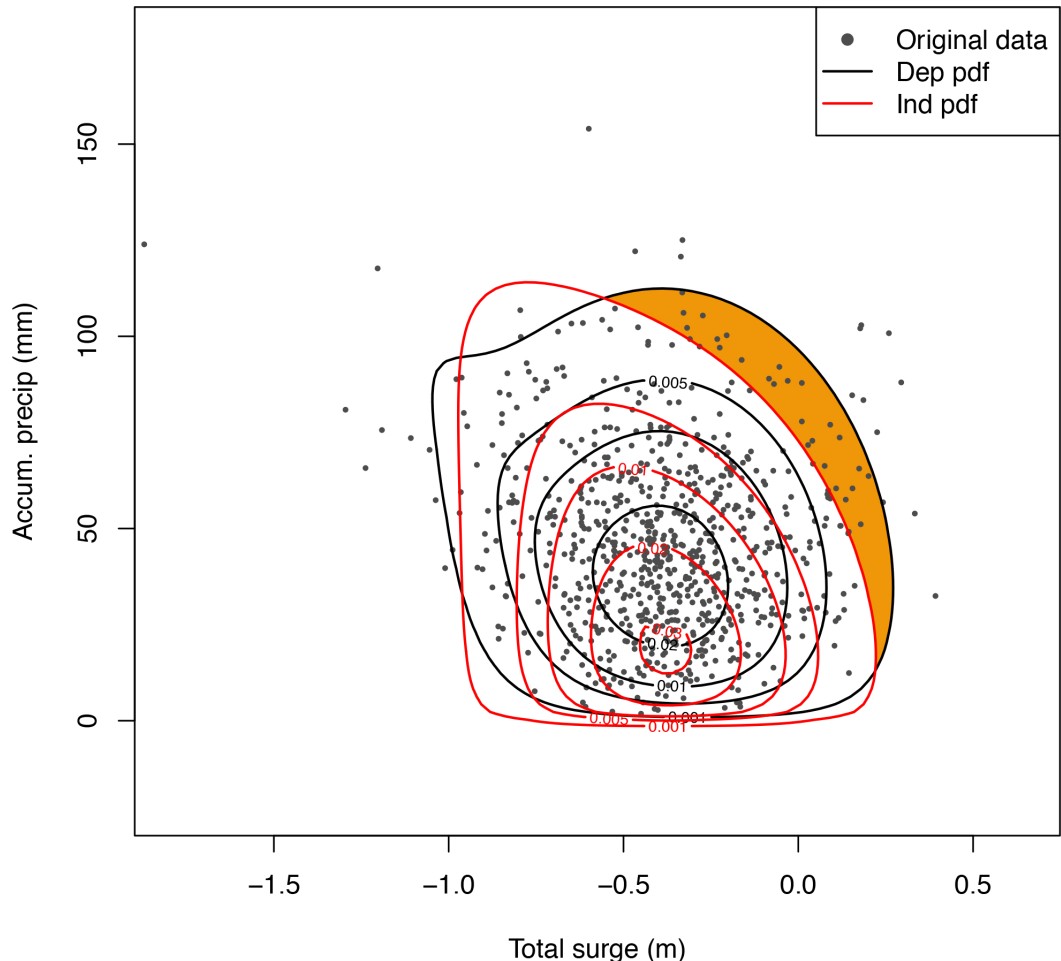

**Figure 4.** Scatter plot of $S_{36h,\min}^T$ and $P_{12d,\text{acum}}$ and its joint PDF corresponding to original data (black) and shuffled data (red). Shaded orange area highlights the increased probability of extreme $S_{36h,\min}^T$ and $P_{12d,\text{acum}}$ for the original data.

mizes the performance of the impact function and hence the performance of the statistical modelling of return level estimates. It is therefore important to distinguish between the correlation/dependence between the selected predictors, and the correlation/dependence between the drivers (although the former informs the latter). There is certainly a number of ways one could define the drivers to better portray such dependence but, regardless of that, when broadly speaking about positive dependence/correlation between drivers one would refer to the increased likelihood of concurrent drivers that contribute to impactful

events, the so-called "compound effects". As shown in Fig. S1, positive compound effects are not necessarily associated with positive values of $\tau$ between the corresponding conditioned predictors. Compound effects can still be investigated by comparison with estimates obtained from shuffled (independent) data, expressed by either $\tau$ or the associated return level estimates (as shown in Section 4.2). For example, the positive dependence between surge and precipitation is not depicted by the plain





correlation between $S^T_{36h,\mathrm{min}}$ and $P_{12d,\mathrm{acum}}$ but by the positive shift between the corresponding correlations obtained for the

original and shuffled data. Moreover, although such dependence has an impact on WL return levels (Section 4.2), the fact that

$\tau$ between $S^T_{36h,\mathrm{min}}$ and $P_{12d,\mathrm{acum}}$ is weak also indicates that the dependence between drivers is not very strong.

### 4.1.2 Seasonal variability

To increase process understanding and strengthen the link between the statistical framework and the physical processes, we

investigate the seasonal variability of the dependence structure between $S^T_{36h,\mathrm{min}}$ and $P_{12d,\mathrm{acum}}$. $\tau$ is lowest during winter (DJF:

-0.13) and increases in spring (MAM: 0.01) and summer (JJA: 0.10) and drops again in the fall (SON: 0). This variability is

caused by the underlying physical factors leading to extreme water levels. In general, surge contributes more to $WL_{\mathrm{max}}$ than

precipitation (see Section 3.3). This is consistent with the correspondence between the monthly frequency of $WL_{\mathrm{max}}$ events

and the monthly frequency of the annual maximum of min total surge over 36-h time windows (without being conditioned to

$WL_{\mathrm{max}}$) (Fig. S6(a) vs S7(b)). In winter surge becomes the most predominant driver, and precipitation has a small contribution

compared to other seasons (particularly the summer). This agrees with the lowest seasonal correlation between $S^T_{36h,\mathrm{min}}$ and

$P_{12d,\mathrm{acum}}$ obtained for this season.

We also investigated separating the statistical analysis into seasonal clusters. It did not lead to an improved representation of

$WL_{\mathrm{max}}$ events and led to a smaller statistical sample. The latter was particularly critical for spring and summer, as the number

of annual maxima events is unevenly spread over the annual cycle and few of these events occur in the warmer seasons. The

majority of $WL_{\mathrm{max}}$ occurs in the fall (Fig. S6(a)) for which the water level is also larger (Fig. S6). Therefore, we continue our

analysis with all-year results and ignore the seasonal signature of WL return levels.

### 4.1.3 Variability as a function of tides

The correlation between surge and precipitation varies as a function of the tide elevation, as shown in Table 3. There is a

tendency of intensified positive dependence between $S^T_{36h,\mathrm{min}}$ and $P_{12d,\mathrm{acum}}$ for higher $T_{12h,\mathrm{min}}$, i.e. for smaller tidal ranges

and higher low tides. This is apparent for both the surge predictor in the 3D case ($S_{72h,\mathrm{mean}}$) and the total surge predictor

($S^T_{36h,\mathrm{min}}$) in the 2D case. This result is in contrast with findings of van den Hurk et al. (2015), who argued that surge and

precipitation had a weaker correlation for most extreme $WL_{\mathrm{max}}$ which they attributed to low tidal range between high and low

tides, as extreme water level tends to occur in neap tide conditions.

Indeed, there is a positive dependence between $T_{12h,\mathrm{min}}$ and $WL_{\mathrm{max}}$ ($\tau$=0.10), which is reflected by a positive shift of the

low tides prior to $WL_{\mathrm{max}}$ with respect to the distribution of all low tides (see Fig. 2(c)). Also, the upper 10% percentile of

$T_{12h,\mathrm{min}}$ occurs in the fall season (Fig. S8), when the largest water level events tend to occur (Fig. S6). This is consistent with

the lower amplitude in the major tidal constituents in September/October in the North Sea (Gräwe et al., 2014).

However, $P_{12d,\mathrm{acum}}$ and particularly $S^T_{36h,\mathrm{min}}$ have a greater impact on $WL_{\mathrm{max}}$ than $T_{12h,\mathrm{min}}$. This is reflected in their

respective rank correlation coefficients: $\tau = 0.23$ ($P_{12d,\mathrm{acum}}$ and $WL_{\mathrm{max}}$) and $\tau = 0.42$ ($S^T_{36h,\mathrm{min}}$ and $WL_{\mathrm{max}}$) ($\tau = 0.36$ for

$S_{72h,\mathrm{mean}}$ and $WL_{\mathrm{max}}$). Also, we argue that it is not evident whether the correlation between surge and precipitation is weaker

for extreme return water levels. The tail of the return level plot is affected by sampling variability. As an example, Fig. S13





**Table 3.** $\tau$ correlation between $S^T_{36h,\min}$ and $P_{12d,\mathrm{acum}}$, and $S_{72h,\mathrm{mean}}$ and $P_{12d,\mathrm{acum}}$, as a function of $T_{12h,\min}$.

| $T_{12h,\min}$ range | $S^T_{36h,\min}$ | $S^T_{72h,\mathrm{mean}}$ |
|---|---|---|
| $T_{12h,\min}$<10th percentile | -0.08 | -0.13 |
| $T_{12h,\min}$<50th percentile | -0.06 | -0.09 |
| $T_{12h,\min}$>50th percentile | -0.02 | -0.02 |
| $T_{12h,\min}$>90th percentile | 0.08 | 0.15 |

illustrates the variation of the range of uncertainty in estimating the 800-year return level by sampling 800 and 100,000 events, respectively, from our statistical framework for both the independent and dependent cases. We empirically obtain that with a single 800-year realization there is a probability of 12% of the 800-year return level from original data to be smaller than the
800-year return level based on the shuffled data. However, when sampling 100,000 events, the probability is virtually zero. This indicates that estimates about the variability of the role of driver dependence on generating high water levels might be subject to sampling uncertainty for return periods of similar value as the sample size length. In any case, clustering by tides reveals that a weaker correlation between $S^T_{36h,\min}$ and $P_{12d,\mathrm{acum}}$ is more likely to happen with lower $T_{12h,\min}$ and therefore larger tidal ranges. Separating the statistical analysis into tidal clusters did not lead to improvement, but we further investigate
the tide effect in the 3D case (see Section 4.2).

### 4.1.4    Climate variability

To assess the effect of the internal variability of the climate system on the correlation between the selected predictors, the correlation between $S^T_{36h,\min}$ and $P_{12d,\mathrm{acum}}$ is estimated for each individual member of the SMILE (50 years per member) (Fig. S9a). The correlation ranges between -0.18 and 0.04 and its mean is -0.05 (equal to the value obtained using 800 years of
data). However, none of these values are statistically significantly different from zero.

     The correlation difference between original and shuffled data (which indicates the positive dependence between surge and precipitation, see Section 4.1.1), is largely affected by climate variability. Fig. S9b-k show the variability of $\tau$ and its statistical significance (at the 95% confidence level) for the shuffled data, which leads to a range of the correlation difference from -0.26 to 0.36 accounting for all ten shuffles. This indicates that internal climate variability has a pronounced impact on the estimation
of compound effects. However, our results are affected by the definition of the predictors, and therefore cannot be generalized. Section 4.2 further investigates this matter in terms of the return levels estimates.

### 4.2    Return water level estimates: compound effects and climate variability

In the following, the proposed statistical framework is validated against the inland return water level estimates provided by van den Hurk et al. (2015), describing results from the marginal and dependence assessments that form the basis of the
methodology presented here. This section also showcases the sensitivity of the three main methodological components (impact function, marginal distributions, and dependence assessment) to the length of data availability and the variability across the different members of the SMILE.





### 4.2.1 Joint probability density function

To estimate the inland water level based on the 2D model, the normal and the Weibull distributions are selected as the optimal
probability distributions to fit the marginals for total surge and precipitation respectively. To represent the joint behavior of
the two selected predictors, the rotated Tawn type I copula is selected. Similarly, in the 3D case a normal distribution fits
both tide and pure surge accurately, and precipitation is well described by a Weibull distribution. The vine structure that most
accurately describes the dependence between these three variables contains the following bivariate copulas: rotated BB1 (270°)
(dependence between $P_{12d,\mathrm{acum}}$ and $T_{12h,\mathrm{min}}$), Frank (dependence between $T_{12h,\mathrm{min}}$ and $S_{72h,\mathrm{mean}}^{T}$), and rotated Clayton(90°)
(dependence between $T_{12h,\mathrm{min}}$ given $S_{72h,\mathrm{mean}}^{T}$, and $P_{12d,\mathrm{acum}}$ given $T_{12h,\mathrm{min}}$). The structure of the regular vine is given in
Fig. S11.

### 4.2.2 Compound effects

To quantify the compound nature of WL, WL return levels are estimated considering independent drivers and used as reference.
Generally, the calculation of return periods for independent drivers can be performed by forcing an independence copula
or by randomly sampling from the fitted marginals directly (Genest and Favre, 2007). However, we selected the predictors
conditioned to $WL_{\max}$ in order to optimize the reproduction of inland water levels calculated by the impact function. This
step affects the correlation between the predictors (see Section 4.1.1 and Fig. S1), which is why zero correlation between surge
and precipitation does not equal to zero correlation between $S_{36h,\mathrm{min}}^{T}$ and $P_{12d,\mathrm{acum}}$. In fact, $\tau$ associated to $S_{36h,\mathrm{min}}^{T}$ and
$P_{12d,\mathrm{acum}}$ obtained from the shuffled data (independent case) features a correlation with a value of -0.15. Hence, our statistical
framework cannot reproduce the return period curves of the shuffled data when using an independent copula to describe the
dependence structure between $S_{36h,\mathrm{min}}^{T}$ and $P_{12d,\mathrm{acum}}$.

To assess the independent case, we use the predictors defined in Table 1 obtained from the shuffled data and we follow
the same procedure as for the dependence case to obtain the corresponding return water levels. Results for both cases are
shown in Fig. 5 (2D case) and Fig. S14 (3D case), where return periods/levels are compared against the empirical estimates
by van den Hurk et al. (2015). Both 2D (Fig. 5) and 3D (Fig. S14) approaches reproduce compounding effects with high skill.
The small difference between these 2D and 3D cases shows that adding complexity to our framework does not necessarily
improve performance. However, the trivariate model is slightly better at reproducing the independent case. Despite overall
good performance, both 2D and 3D approaches differ slightly from the empirical data for the highest return periods. However,
as noted in Section 4.1.3., the tail of the return plot is sensitive to the number of simulations used to obtain such estimates (see
Fig. S13). This explains the disagreement between the modelled and the empirical estimates for large return periods (modelled
lines are more parallel than empirically estimated lines), as we obtained these curves by simulating larger samples than the
empirical analysis (100,000 events).





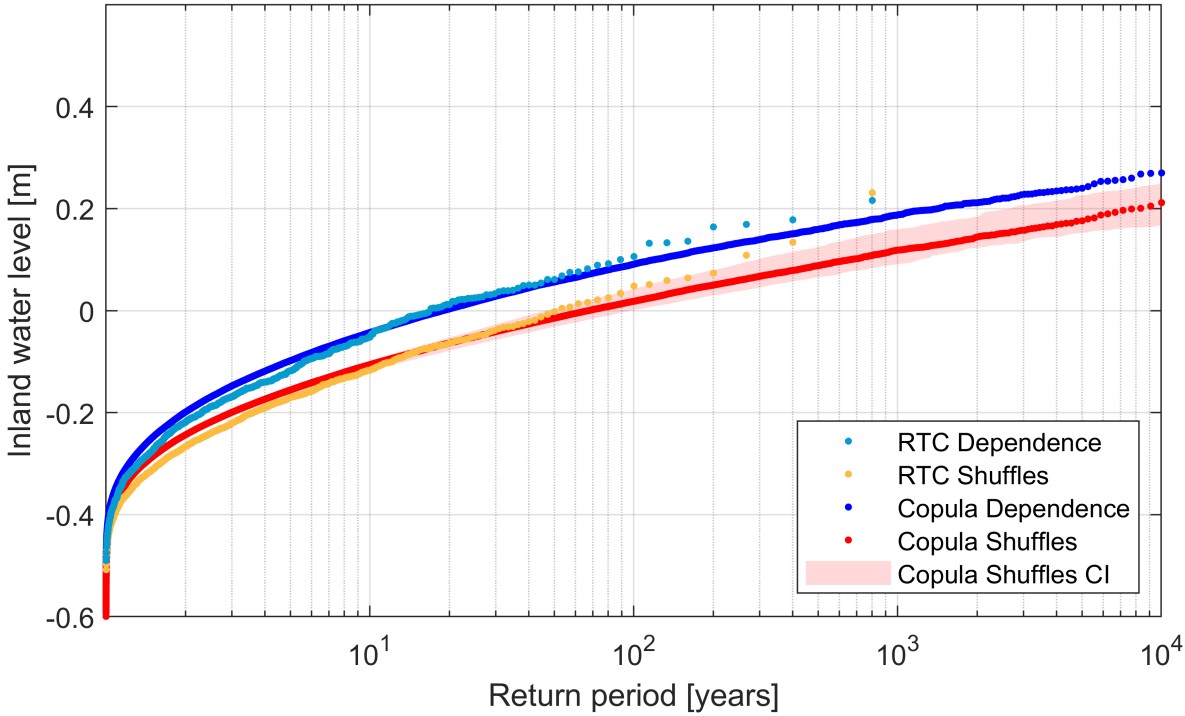

**Figure 5.** Inland water level return level against estimated return period using a bivariate copula model (2D case). Blue and red dotted lines depict the dependence and independence case, respectively. Transparent red denotes confidence intervals, which account for the uncertainty range between the $5^{th}$ and $95^{th}$ percentiles, as computed from all shuffles. Light blue and orange dots represent the return values empirically obtained by van den Hurk et al. (2015).

### 4.2.3 Climate variability

In Section 4.1.4 we show the effect of the climate variability on the predictors' dependence structure. Here, we explore the

effect of climate variability on each component of our statistical framework: the impact function, the marginal distribution, and the copula function. In particular, we investigate the impact on (i) the estimates of WL return levels corresponding to the dependence case (Fig. 6) and (ii) the ratio of the estimated return periods from the shuffled predictors ($RP_s$) to those derived by accounting for dependence between predictors ($RP_d$) (Fig. 7). This ratio indicates the bias in return period calculation if dependence between drivers was ignored and is used as a proxy of the compound effects, i.e. the increased probability of

extreme WL due to the positive dependence between surge and precipitation. Table 4 specifies the settings used to produce Fig. 6 and 7.

**Figure 6.** Return water level against estimated return period using a bivariate copula. Blue dots depict the return level estimates obtained using the proposed statistical framework (using 800 years of data). Green illustrates the uncertainty associated to internal climate variability, represented by bounds computed using the 5th and 95th percentiles from all 50-year ensembles, and the median value (dots). This is assessed for each component of the methodology: a) 50-year ensembles are used for all components; b) same as a) but impact function is trained with 800 years of data; c) same as b) but using bin sampling approach; c) 50-year runs are used for copula fitting only; d) 50-year runs are used for total surge marginal fitting only; and e) 50-year runs are used for precipitation marginal fitting only (see Table 4).

**Figure 7.** Compound effect (estimated as ratio between return periods as obtained from shuffled and original data) against return water level using a bivariate copula. Blue dots depict the values obtained using the proposed statistical framework (using 800 years of data). Green illustrates the uncertainty associated to internal climate variability, represented by bounds computed using the 5[th] and 95[th] percentiles from all 50-year ensembles, and the median value (dots). This is assessed for each component of the methodology: a) 50-year ensembles are used for all components; b) same as a) but impact function is trained with 800 years of data; c) same as b) but using bin sampling approach; c) 50-year runs are used for copula fitting only; d) 50-year runs are used for total surge marginal fitting only; and e) 50-year runs are used for precipitation marginal fitting only (see Table 4).





**Table 4.** Settings used in subpanels of Figures 6 and 7 to assess climate variability (green).

| Subpanels | 50-year runs | | | | 800-year ensemble | | | |
|---|---|---|---|---|---|---|---|---|
| | Impact function | Copula | Total surge PDF | Precipitation PDF | Impact function | Copula | Total surge PDF | Precipitation PDF |
| a | x | x | x | x | | | | |
| b | | x | x | x | x[*] | | | |
| c | | x | x | x | x | | | |
| d | x | | | x | x | | x | x |
| e | | | x | | x | | x | x |
| f | | | | x | x | x | x | |

[*] Impact function is not optimally trained, i.e. bin sampling approach is not implemented.

First, Fig. 6(a) shows WL return period and level estimates for the bivariate case, and associated variability computed from all subsets of 50 years for each component. Large uncertainty intervals surround the average of values based on these 50-year subsets, and this average return period curve is shifted downwards compared to the 800 year reference curve approach. The

general tendency of the regression model to underestimate return levels, especially for high return periods, is mainly caused by the fact that we cannot perform the bin-sampling approach with only 50 years of data. Indeed, not performing the bin-sampling procedure when using the entire dataset (800 years of data) also leads to an underestimation of return values for both dependent and independent cases (Fig. 6(b)). The optimal training of the impact function by means of bin sampling eliminates the tendency to underestimate high return periods, as shown in Fig. 6(c) where the proposed function in Subsection 3.3 is

applied while using 50-year ensembles for marginal and copula fitting. Yet, uncertainty is not reduced, which illustrates that most uncertainty related to internal climate variability is introduced by other framework components. Similar to Fig. 6(a) and (c), Fig. 7(a) and (c) show the variability of the return period ratio when 50-year ensembles are used for all framework components and when the impact function is optimally trained, respectively. Return period ratios are likely to vary significantly when only 50 years of data are available as noted by the large green intervals (Fig. 7(a) and (c)). Furthermore, there is a tendency

to underestimate compounding effects even when the impact function has been optimally trained (Fig. 7(c)).

Second, the effect of climate variability on copula fitting and its impact on inland WL return level estimation are shown in Fig. 6(d). Here, we apply the optimally trained impact function and use the entire dataset to fit the marginals while varying the length of the data used in copula fitting. As expected, the copula fitting does not generate significant differences between the 50-year runs as $\tau$ becomes virtually zero for all 50-year runs (see Section 4.1.4, Fig. S9(a)). This low variability induced

by copula fitting, however, does not imply that bivariate copula models are generally unaffected by climate variability. In this study, copulas do not play a significant role in the estimation of inland WL return period estimation for the 2D dependence case. While there is dependence among drivers, the Kendal's $\tau$ for the 800 years of the selected (conditioned) predictors is very close to zero. Hence, shortening the dataset length does not affect the reliable estimation of WL in terms of copula modelling for the dependence 2D case. Nonetheless, climate variability does affect the estimation of WL for the shuffled data (not shown)

due to the inherent variability in the corresponding $\tau$ and copula fitting (Fig. S9b-k)). This suggests that the use of short records probably affects the estimation of compound effects. Indeed, Fig. 7(d) clearly illustrates that the use of short records tends to lead to an underestimation of compound effects. Climate variability also causes a large uncertainty of return period ratios.





Third, to explore the effect of climate variability on marginal fitting, we tested and fitted different suitable probability distributions to the marginals of all 50-year ensembles, while using 800 years for copula fitting and the optimally trained impact

function to transform simulations. A comparison between Fig. 6(e), Fig. 7(e), Fig. 6(f) and Fig. 7(f) shows the uncertainty associated to total surge and precipitation data marginal fitting. We find that most uncertainty in estimating WL return levels is associated to the fitting of the total surge distribution (Fig. S10(a)). This uncertainty is reflected in the water level estimates, since the total surge is the predominant driver. Furthermore, comparing Fig. 7(d-f) reveals that the tendency to underestimate compounding effects in Fig. 7(d) is mainly introduced by the copula fitting. Hence, short records might prohibit a proper

estimation of compound effects due to poor copula fitting.

An analogous uncertainty analysis was performed for the trivariate case (Fig. S14), examining the uncertainty associated to each component of the proposed statistical framework. Although generally similar insights were obtained as for the bivariate uncertainty assessment, some differences are worth mentioning. For instance, copula fitting (Fig. S14(c)) presents larger uncertainty intervals than for the bivariate case. As the predictors are defined differently in the trivariate case, the correlation

between them has also changed and has become crucial to reproduce WL dependence curves. In addition, separating total surge into surge and tidal range reveals that marginal fitting uncertainty is mostly caused by surge, followed by tides (see Fig. S10(c) and (d)). Although tidal range is an important factor determining the occurrence of extreme WL in our study case, the surge is the most important variable explaining the behavior of inland WL (as seen in Section 3.3).

In sum, we find that the internal variability of the climate system represented by the variability between the 16 50-year

members induces a large uncertainty range at every step of our statistical framework. The impact function cannot be properly calibrated with 50-year data. Furthermore, compound effects tend to be underestimated when applying short records to fit the copula.

## 5 Conclusions

In this study we developed a copula-based multivariate statistical framework that produces robust estimates of compound

extreme inland water return levels for a highly managed reservoir in the Netherlands. This work was motivated by a near-flooding event in 2012, which was empirically analyzed by van den Hurk et al. (2015) based on a single model initial-condition large ensemble (SMILE) consisting of a set of 16 50-year simulations. Like in van den Hurk et al. (2015), we used these 16 members as 800 years of current climate conditions that account for the internal variability of the climate system. In particular, we defined simulations of the inland water level as the impact variable, and total surge and precipitation as the underlying

drivers. To assess compounding effects, we used a randomized ensemble of independent drivers which van den Hurk et al. (2015) obtained by shuffling the 50-year runs, thereby removing the correlation between surge and precipitation but preserving their climatological characteristics.

The high degree of human management in the system studied poses a challenge to select suitable predictors and subsequently developing an impact function that is skillful at predicting water levels as a function of the underlying drivers. We considered

bivariate and trivariate models (which was implemented after separating total surge in surge and tidal ranges) but the latter



did not lead to overall improvement. Optimal predictors were found after an iterative process to optimize the performance of the impact function and return level estimates. After testing several options, we defined the annual maximum water level as predictand, and the 12-day cumulative precipitation and 36-h minimum total surge prior to $WL_{\max}$ as predictors. The resulting optimal impact function is a multilinear regression model with a bin-sampling approach that gives more weight to the most

extreme water level events in the calibration process. Total surge is found to be the predominant driver.

    Our statistical model shows that, although not very strong, the dependence structure between drivers (surge and precipitation) contributes to increased return water levels, as was found empirically by van den Hurk et al. (2015). However, due to the conditioning of the proposed predictors on the impact variable this is not reflected in a positive $\tau$ between the selected predictors, but the positive dependence is implicitly assessed by comparing the joint probability distributions and return level

estimates to results obtained from the shuffled (independent) data. Some extreme water levels are primarily driven by surge (especially those occurring in winter) but compoundess increases for other seasons. A copula-based multivariate statistical framework is generally able to capture the complex compound nature of precipitation and surge, and to reproduce extreme inland water return levels at the local scale, also under conditions where the strong management of the hydrological system was not explicitly represented in the underlying data.

Furthermore, we performed a unique uncertainty assessment to explore the impact of internal climate variability on the return water level estimates. The use of a subset of 50-years of data (which is the typical record length available from observed records) was tested for different components of our framework, namely the impact function, the copula fitting, and the marginal fitting. Using a degraded impact function training leads to a consistent underestimation of the return levels, as the bin sampling approach is not feasible for 50 years of data. The marginal fitting of total surge is the factor that most contributes to uncertainty

of the return level estimates. For the 2D case, copula fitting does not lead to additional uncertainty and shortening records does not significantly impact the return level estimates. However, low variability provided by copula models is due to their insignificant role in the estimation of WL return level for the dependence 2D case, as correlation between the selected predictors (conditioned to WL) is close to zero. Indeed, the 2D case could be simplified with an independent copula with no major impact on return level estimates. Yet, dependence models are still crucial to reproduce and understand compounding effects, as the

dependence structure does play a significant role when modelling the shuffled data. The use of the 50-year subset leads to a tendency to underestimate the increased probability of extreme WL due to inherent positive dependence between surge and precipitation. For the 3D case, increased dependence between the predictors and a larger model complexity leads to increased uncertainty induced by copula fitting when shorter records are used. We emphasize that these findings are highly case-specific and dependent on the chosen statistical framework. However, this case study illustrates that internal variability is a major source

of uncertainty for estimation of extreme inland water levels and the associated compound effects.

    We conclude that our statistical framework needs larger sample sizes than we would typically obtain from observational data in order to reproduce accurate extreme inland water level statistics. Observational time series are one possible realization of the climate system within its boundaries of internal variability. Therefore, short records present challenges to properly estimate the relationship between predictors and predictand, marginal distributions and dependence patterns. Large sample sizes made





available from the application of SMILEs are valuable to investigate compound events and the associated uncertainties induced
by internal variability.

*Data availability.* The SMILE data used are identical to the dataset used van den Hurk et al. (2015), and are not made publicly accessible
due to the large volume and associated cost for a (semi-)permanent repository. Any reasonable request for access to SMILE data can be
addressed to B.vH. Post-processed quantities used for the analysis described in this paper are available at https://github.com/victor-malagon/
CF_theNetherlands_data, DOI:zenodo.org/record/4088763.

*Author contributions.* V.M.S., M.C.-P. and B.P. led the analysis and development of the multivariate statistical model, and writing of the
manuscript. B.vH. and E.R. conceived the experiment design, co-supervised the project and contributed to writing. M.C.-P. co-supervised
the project and contributed with experiment design. B.P. co-supervised the research project. Z.H., T.K., L .Z. and N.H. contributed with data
analysis and proofreading.

*Competing interests.* We have not competing interests

*Acknowledgements.* This project research was developed in the context of the "Institute of Advanced Studies in Climate Extremes and Risk
Management" (October 2019, Nanjiing, China) and "the Damocles training school on statistical modelling of compound events" (September
2019, Como, Italy). We thank the corresponding organizers and sponsors. The Institute of Advanced Studies was organized by the World
Climate Research Program (WCRP), led by the WCRP Grand Challenge on Weather and Climate Extremes (GC-Extremes), in collaboration
with Future Earth, Integrated Research on Disaster Risk (IRDR) and Nanjing University of Science and Technology (NUIST). This activity
was endorsed by the International Science Council (ISC). Damocles is supported by the European COST Action CA17109 within the EU
Framework Programme Horizon 2020. We are grateful to Erik van Meijgaard and Klaas-Jan van Heeringen for making available the RCM
and RTC-Tools simulations. E.R was funded by the European Union's Horizon 2020 research and innovation programme under the Marie
Sklodowska-Curie grant agreement No 707404. L.Z. was funded by the National Key R & D Program of China (2017YFA0603804) and
China Meteorological Administration Special Public Welfare Research Fund (GYHY201306024).





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
