# Peer review of "Multivariate statistical modelling of extreme coastal water levels and the effect of climate variability: a case study in the Netherlands"

_Hydrology and Earth System Sciences, 2020_

## Referee Comment (RC1) · Anonymous Referee #1 · 23 Nov 2020

The manuscript 'Multivariate statistical modelling of extreme coastal water levels and the effect of climate variability: a case study in the Netherlands' assesses an impact function that can reproduce inland water levels in a human controlled system by event sampling and conditioning the drivers. By modelling the dependence structure between the different drivers to generate paired synthetic events, the authors are able to assess compounding effects of surge and precipitation on inland water levels. Overall, this study is an interesting read and I commend the authors for their nice work. It uses well-established methods and builds on previous assessments. Furthermore, it provides new insights in modelling compounding effects of surge and precipitation, and an interesting analysis of climate variability and using a short subset of the data. The manuscript also provides interesting and detailed information and discussion on underlying processes of the predictor selection and interpretation of the compounding effects of the two drivers. However, in its current form this study has a number of limitations that I would like to see addressed. For instance, the contextualization of using a case study in an area with a high degree of human management is lacking, steps undertaken in the methods section need more clarification, and decisions undertaken in the results need more clarification and transparency. Therefore, I propose to reconsider this manuscript for publication upon revision of the following issues.

Specific comments:

1. The title of this manuscript is framed as a case study that provides a statistical framework for assessing extreme coastal water levels and climate variability that can be used for other case studies as well. By framing the title like this I would expect a discussion in the manuscript that addresses how this statistical framework (e.g. conditioning the drivers) can be used for other areas of interest or even a different region in the Netherlands. This contextualization of how a user can use this framework in other areas of interest is lacking in the manuscript's current form.

2. The case study in the Netherlands provides an analysis on an area with a high degree of human management. As the title of the manuscript does not cover this, I would suggest to either add this information to the title or add a short discussion on how this statistical framework can be used for other areas which do not have a high degree of human management.

3. Throughout the manuscript, water levels are most often referred to as inland water levels (line 1), however sometimes the authors use solely water levels without the adjective 'inland' (e.g. line 99), or extreme coastal water levels as is stated in the title of the manuscript. I would suggest to stay consistent with the terminology and provide a clear description of the water levels (e.g. how much inland, coastal/inland water levels).

4. While the manuscript discusses relevant previous studies in the introduction (line 59-73), the research gap is not pointed out clearly. As a consequence, the novel aspects of this study and the research gap do not come across strongly. Therefore, I suggest adding more detail to the this section in the introduction.

5. In order to improve readability, I would suggest to rephrase line 116 by using frequencies, i.e. more frequent in original data or less frequent in shuffled data.

6. In section 2, data and study area, please provide more background information how the predictors total surge and precipitation were derived. For instance, information how the surge and tide are added (van der Hurk et al., 2015).

7. In line 137-138, you mention that the performance of the impact function is highly sensitive to the selection of the predictors, yet no sensitivity analysis or the degree of sensitivity is reported or shown. Please provide more information and details on how sensitive it is.

8. Please contextualize, if possible, why the annual maxima surge events are at least 3 days (5th percentile), see Figure 1b and lines 174-176.

9. In lines 176-178, please provide more contextualization about the tradeoff and why the minimum total surge is selected.
10. In Table1, the selected predictors of the two cases are reported, taking into account the three aspects mentioned in lines 141-145. Additionally, information is provided for the selection of predictors in lines 159-161. Please provide more information about the optimization technique used. Why was the maximum (next to the minimum and mean) for the conditioning not included? Which approach was used for this during conditioning of the drivers (MLR, MLRbin, ANN, etc.)? Is the performance of the predictor selections evaluated on the metrics used throughout this study, or the tradeoff between the metrics and visual inspection of the events that exceed the flood warning level as in line 222-223? What is the time step of additional hours prior to the event used for this selection? Were all possible combinations of the selected time steps and statistics evaluated or was an optimization technique used for this (e.g. random search)?
11. In line 191-193, please provide short details on which architecture and hyper parameters are used for the machine learning approaches.
12. Like equation 2, are the predictors in equation 3 for the 3D case also standardized?
13. In lines 250-253, please provide more information on what terms the 3D case generally does not outperform the 2D case. To me it seems that the 3D case performs better on the reported metrics. Above the flood warning level, the differences looks only marginal (confidence interval of the 1000 bootstrap runs not reported). If the focus of this manuscript is on the distribution of extreme cases above the flood warning level, then it should be clearly stated in the manuscript. Additionally, lines 366-367 report that adding complexities does not necessarily improve performance. However, the reported metrics show an improvement. In lines 367-368 the authors report that the performance between the two cases differ slightly for higher return periods. Why did you choose to not report metrics (e.g. MAE) of those extremes of extreme events? Moreover, lines 440-441 report that the 3D case did not lead to an overall improvement. Pleas provide more information to the respective section why those decisions are taken and on what basis (e.g. define overall in overall improvement).
14. Line 291-292, please provide more information or give possible examples on the underlying physical processes
15. In line 297-298, you mention that separating the analysis in seasonal clusters did not lead to an improvement, but do not report to what extent. Please provide more information to the respective section. Additionally, in line 324 you mention that separating the statistical analysis in tidal clusters did not lead to an improvement. Please specify to what it did not lead to an improvement.
16. The section about seasonal variability evaluates the dependence structure of the predictors and reports the Kendall's rank correlation for the respective seasons. This is a very interesting read and discussion, however the authors report in line 298-299 that the spread of annual maxima events is uneven and that for some months few events occur. Have the authors considered restructuring the inland water levels maxima in seasonal maxima, resulting in 800 maxima inland water levels per season?
17. Please provide contextualization on the results reported in lines 327-330.

Technical corrections:

- In line 124-132, it would improve readability to also refer to the respective sections in the methods for the different steps of the conceptual model.
- In line 200-202, extreme water levels exceeding 0 meter is used to describe the higher end of the water levels, however it would be more sensible for the reader at this stage to refer to in percentiles or flood warning level as indicated in the sup.

- In line 260, do you mean 'inland' water level?
- Line 338 misses a word: 'in the following …'.
- Line 366 Fig S14 should be Fig S12
- Line 372 now reads as if empirical analysis consists of 100,000 events. Please rephrase.
- In the caption of Fig 6, transparent should be added to 'Green illustrates the uncertainty …'
- In the caption of Figs 6 and 7 c, d, and e don't match up.
- Table 4 subpanel e, to my understanding the copula of 800-year ensemble should be marked and not the total surge of 800-year ensemble.
- Please clarify the sentence in line 414-415 starting from 'hence'.
- In the supplementary information line 45 contains a duplicate of 'a'.
- In the supplementary information lines 51-52 reports difference between shuffled dataset and dependent dataset while using the same symbols. As a suggestion the authors can use for example $A_{C_{max,shuffled}}$ in order to increase readability.
- The caption of Fig S7 contains a duplicate of 'that'.

---

## Referee Comment (RC2) · Anonymous Referee #2 · 11 Dec 2020

The manuscript titled 'Multivariate statistical modelling of extreme coastal water levels and the effect of climate variability: a case study in the Netherlands' presents an interesting attempt to quantify the joint probability of coastal water levels and precipitation that ultimately create a compound flood hazard. They use 800 synthetic annual maxima events to define the marginals of copulas and create a trained impact function to relate predictands and predictors. The degree of uncertainty introduced by shorter records is also quantified as a commentary on the importance of data volume for such methodologies. While I do think the publication will ultimately be a quality contribution to the literature, it currently is vague on some methodology components that need further clarification. I recommend that the manuscript be returned to address the following thoughts:

- In general, there are a lot of references to figures in the supplemental information that feel as if they are written in the same manner that one would normally refer to an in-text figure. If showing these figures are crucial to communicating the results, then I feel they should be in the main paper. Otherwise I suggest rewriting the sections (i.e. 4.1.2, 4.1.3, 4.1.4, etc.) to explain the results in words without referencing a take-away point that a reader would need to see a figure to understand. You can then tell the reader that further information is available in the supplement.
- The introductory paragraph refers to the same author/lab groups efforts in 5 straight individual sentences. While subsequent paragraphs show the author's have a broad grasp on literature beyond this one lineage, I recommend broadening the background to highlight that the motivation for this work does not arise simply from one group's efforts. There are many other works that have identified and attempted to account for multivariate climate drivers of compounding events (e.g. Anderson et al. 2019, Serafin et al. 2014, Rueda et al. 2016).
- I'll admit I am confused by the tidal variability included in Figure 1. The text at Line 114 indicates that the tide cycle is added but doesn't give any specifics (I suggest adding these specifics to improve transparency). Figure 1 makes it look like all 800 events had the maximum occur at the same phase of the tide? Otherwise the bold tidal level would be a more flat line with a large envelope of variability around it? If the events do all occur at the same tidal phase then that would be a significant limitation of this work. Perhaps Figure 1 is only a single example taken from the 800 annual and the text caption for the Figure could be rewritten to prevent the interpretation that it is derived from all 800 scenarios.
- Are copulas fit to purely empirical distributions? At which point the underlying assumption is that the 800 events can accurately represent the tails of the distributions? If this is the assumption being made then I think it should be explicitly stated in the manuscript and acknowledged as a potential limitation for obtaining extremes.
- Although a paragraph at the beginning of Section 2 does describe the study site, I think an annotated figure of the coast, the physical point where all data is obtained, and the square area or arial footprint of the watershed catching the precipitation could aid the

manuscript. I was left wondering about the coastal configuration, proximity to open water, and proximity to human altered landscapes.

- I think the paragraph between lines 112-120 could grow to be multiple paragraphs that detail the methodology from van den Hurk et al. (2015), as this manuscript is heavily dependent on that work.
- Although the explanation of copulas is suitable for publication, I think that the author's dynamical interpretation of the final copulas could be useful. By that I mean, why does a Frank copula fit better and what does that tell us about the dynamics of the compound hazard?
- I think the usefulness of this case study to readers may be improved by including a commentary on what physical processes are or are not being wrapped up into the relatively broad predictors. Does the original modeling framework exhibit sea level anomalies at longer frequencies than just meteorological surges and tides (i.e. monthly or seasonal anomalies)? Does the location of this virtual tide gauge experience waves? Perhaps a paragraph at the end of the discussion could address limitations and how extensible the study is to other sites.

---

## Author Comment (AC1) · 19 Jan 2021

We, the authors, thank the anonymous reviewer for providing a thorough and comprehensive review. We acknowledge their suggestions will help us improve the quality of the manuscript and hope the suggested amendments satisfy their concerns. Below we provide our point-by-point response to their comments.

1. The title of this manuscript is framed as a case study that provides a statistical framework for assessing extreme coastal water levels and climate variability that can be used for other case studies as well. By framing the title like this I would expect a discussion in the manuscript that addresses how this statistical framework (e.g. condi-

[Figure]

tioning the drivers) can be used for other areas of interest or even a different region in the Netherlands. This contextualization of how a user can use this framework in other areas of interest is lacking in the manuscript's current form.

The statistical framework can be used to other areas of interest, given the availability of relatively long overlapping records of flooding drivers and impact variable. Defining appropriate impact-based predictors that optimize the performance of the impact functions depends on the hydrological characteristics and management of a given system. For systems with low to no management, we would expect a more straightforward construction of an impact function. In any case, composite (average) plots can guide this process where appropriate lags between drivers and impacts should be accounted for. We will add a paragraph in the revised manuscript to discuss this transferability as well as limitations. This study did not include wave-driven water levels (i.e. wave set up). This is a reasonable assumption in the shallow Wadden Sea (sheltered by barrier islands), where surge is the main flooding marine driver. In other locations, the wave contribution, or other drivers such as snow melt, might need to be considered as well. Additionally, we did not account for low-frequency variations of water levels such as sea level rise, which would need to be considered if results where to be extrapolated into the future.

2. The case study in the Netherlands provides an analysis on an area with a high degree of human management. As the title of the manuscript does not cover this, I would suggest to either add this information to the title or add a short discussion on how this statistical framework can be used for other areas which do not have a high degree of human management.

This information will be added to the title and will also be discussed in the main text. A new tentative title is "Statistical modelling and climate variability of compound surge and precipitation events in a managed water system: a case study in the Netherlands".

3. Throughout the manuscript, water levels are most often referred to as inland water

levels (line 1), however sometimes the authors use solely water levels without the adjective 'inland' (e.g. line 99), or extreme coastal water levels as is stated in the title of the manuscript. I would suggest to stay consistent with the terminology and provide a clear description of the water levels (e.g. how much inland, coastal/inland water levels).

We agree with the reviewer about the ambiguity of our previously used terminology. We will modify the terminology throughout the paper in the following manner: coastal water level, to refer to water elevation on the coastal side; and inland water level, to refer to reservoir levels. We hope it is clearer now.

4. While the manuscript discusses relevant previous studies in the introduction (line 59-73), the research gap is not pointed out clearly. As a consequence, the novel aspects of this study and the research gap do not come across strongly. Therefore, I suggest adding more detail to the this section in the introduction.

The last paragraph of the Introduction mentioned the novel aspects of our study. However, to emphasize the novel aspects of our study in relation to existing gaps, we will add more details in the last paragraph of the Introduction, as suggested by the reviewer. We will also change the title including two novel aspects: 1) compound analysis in a managed water system, 2) sensitivity of such analysis to climate variability. Another novel aspect of our study, which will be also explained in the introduction, is the analysis and interpretation of the correlation coefficient of impact-based predictors.

5. In order to improve readability, I would suggest to rephrase line 116 by using frequencies, i.e. more frequent in original data or less frequent in shuffled data.

We prefer to keep the term return period rather than frequencies, as it is a well-known term that is used in many contexts: risk analysis, impact assessment, infrastructure design, etc. However, we will rephrase the paragraph to improve readability.

6. In section 2, data and study area, please provide more background information how the predictors total surge and precipitation were derived. For instance, information how

the surge and tide are added (van der Hurk et al., 2015).

We will provide more information about how total surge (coastal water level in the revised manuscript) and precipitation were derived in Section 2. In a nutshell, surge is calculated from wind speed via an empirical equation that was previously calibrated for the study area. The tidal time series is an artificial extension of the standard astronomical equations and was calculated using all known current tidal constituents for a complete period of 800 years. Total water level is the sum of surge and tide.

7. In line 137-138, you mention that the performance of the impact function is highly sensitive to the selection of the predictors, yet no sensitivity analysis or the degree of sensitivity is reported or shown. Please provide more information and details on how sensitive it is.

We will add a few examples in the Supplementary Material to illustrate the sensitivity of the impact function performance to the selection of predictors. To do so, we will generate some plots assessing the performance of the impact function for different predictor choices (for example: Min Surge 12, 24, 36, 48, 60h, CumPrcp 5d, 12d; Max Surge 36h, Mean Surge 36h, etc.). The inland water level in this specific location is more sensitive to variations in storm surge than to variations of precipitation. Hence, the sensitivity analysis will focus on storm surge.

8. Please contextualize, if possible, why the annual maxima surge events are at least 3 days (5th percentile), see Figure 1b and lines 174-176.

It is reasonable to assume that the relevant duration of storm surge is 3 days as three days is the mean duration of cyclones over East-Central Europe (Bartoszek, 2017). Note that from Figure 1b we cannot conclude however that events linked to annual maxima surge events are at least 3 days long. The shaded area depicts values from 5th to 95th for each time considered. However, storm surge time series of individual events are not necessarily parallel to the mean storm surge, or lower/upper envelopes. As the lower envelope of storm surges is not necessarily linked to a single event, the

associated duration below 3 days does not necessarily have a probability below 5 %.

Reference: Bartoszek, K. (2017) The main characteristics of atmospheric circulation over East-Central Europe from 1871 to 2010. Meteorology and Atmospheric Physics, 129, 113-129.

9. In lines 176-178, please provide more contextualization about the tradeoff and why the minimum total surge is selected.

We will rephrase the text to avoid confusion. Our case study is a water-managed system. To try to prevent inland water levels from reaching extreme levels, the system is regulated by opening the gates around low tide. However, if the coastal water level at the low tide is too high, gates cannot open, water cannot be released, and inland water level might increase due to precipitation and river runoff. Therefore, the annual maximum water level better relates to the previous local minimum of coastal water level (i.e. total storm surge), rather the local maximum, as shown in the composite plot (Fig 2). This differs from a natural water system in which the maximum or mean storm surge would probably be a better predictor to describe extreme water levels. In fact, when the tide and surge are separated, we found that the mean surge (and not the minimum surge) is a better predictor. The choice of 36 h is not randomly selected as a value in between 72h and 12h (from the 3D marine predictors) but is the result of optimizing the impact function performance, for which a wide range of time lags were tested.

10. In Table1, the selected predictors of the two cases are reported, taking into account the three aspects mentioned in lines 141-145. Additionally, information is provided for the selection of predictors in lines 159-161. Please provide more information about the optimization technique used. Why was the maximum (next to the minimum and mean) for the conditioning not included? Which approach was used for this during conditioning of the drivers (MLR, MLRbin, ANN, etc.)? Is the performance of the predictor selections evaluated on the metrics used throughout this study, or the tradeoff between the metrics and visual inspection of the events that exceed the flood warning level as in line 222-

223? What is the time step of additional hours prior to the event used for this selection? Were all possible combinations of the selected time steps and statistics evaluated or was an optimization technique used for this (e.g. random search)?

We will add more details of the performance assessment and optimization. We tested a wide range of predictors, mean, max and min values, for different time lags. This selection was guided by the composite plots and physical understanding of the water system. The revised manuscript will provide some representative examples.

11. In line 191-193, please provide short details on which architecture and hyper parameters are used for the machine learning approaches.

The learning process of the artificial neural network used here is based on stochastic gradient descent, and the activation function is the sigmoid function. The architecture of the network is as follows: input layer with 2 (2D case) or 3 (3D case) neurons; 2 hidden layers with 8 neurons each, output layer with 1 neuron.

The number of trees in the random forest was set to 50, after performing a sensitivity analysis assessing the overall performance of the approach (estimated as root-mean-square error (RMSE) via k-fold validation approach) depending on the number of trees. We selected 50 because increasing the number of trees beyond that value did not lead to an increase in performance.

We refrained from using additional (more sophisticated) machine learning approaches and testing other architectures because we achieved a reasonably good performance using regression for most return periods by means of bin-sampling. We believe this approach is easier to implement, which can aid transferability. We will include this information in the manuscript.

12. Like equation 2, are the predictors in equation 3 for the 3D case also standardized?

No, equation 3 for 3D case is not standardized. The standardization was only shown in Eq. 2 to preliminary demonstrate the importance of total surge (coastal water level in

the revised manuscript) to drive extreme reservoir levels, as compared to precipitation.

13. In lines 250-253, please provide more information on what terms the 3D case generally does not outperform the 2D case. To me it seems that the 3D case performs better on the reported metrics. Above the flood warning level, the differences looks only marginal (confidence interval of the 1000 bootstrap runs not reported). If the focus of this manuscript is on the distribution of extreme cases above the flood warning level, then it should be clearly stated in the manuscript. Additionally, lines 366-367 report that adding complexities does not necessarily improve performance. However, the reported metrics show an improvement. In lines 367-368 the authors report that the performance between the two cases differ slightly for higher return periods. Why did you choose to not report metrics (e.g. MAE) of those extremes of extreme events? Moreover, lines 440-441 report that the 3D case did not lead to an overall improvement. Pleas provide more information to the respective section why those decisions are taken and on what basis (e.g. define overall in overall improvement).

The 3D case performs better based on the reported metrics for the impact function, but the increase in performance compared to the 2D case is minimal so we opted to showcase the simpler case. The focus of the manuscript is on the distribution of extremes but not necessarily above the warning level, as very few events are available exceeding the warning level and this leads to high sensitivity in return period estimates. We will rephrase these results to make our interpretations clearer.

In our manuscript, lines 366-367 report that adding complexities does not necessarily improve performance based on a visual inspection of Figure 5 and Figure S12, instead of reporting metrics. By doing so, we refer to performance when reproducing the return period curves from van den Hurk et al. 2015, i.e., we are not only referring to performance of impact function (where the metrics the reviewer is referring to are reported) but to the entire multivariate statistical framework. In lines 367-368 we also report differences in performance based on a visual inspection, and our decisions to report not overall improvement when using a 3D case are performed on the same basis. We

chose this way of reporting findings as we expected small differences in the metrics and opted to shorten these paragraphs by doing so. We agree with the reviewer that reporting appropriate metrics will help clarify our findings. By means of RMSE, calculated by comparing empirical estimates from van den Hurk et al. 2015 and our equivalent estimates (i.e., taken estimates of WL associated to the same return periods), we do acknowledge a small overall improvement in the 3D model: while the RMSE for the 2D case are 0.0202 and 0.0202 m (dependence and shuffled, respectively), the RMSE obtained in the 3D approach are 0.0189 and 0.0187 m. Regarding performance at the higher return periods, we feel reporting metrics could be highly deceiving, as the empirical estimates above the warning level are not enough no ensure stable statistics and, as reported in our manuscript, there is high sensitivity in the tails, which depends on the number of the available events. We will include the overall metrics and edit the paragraphs containing this information in the manuscript. We hope we addressed the Reviewer's comment properly.

14. Line 291-292, please provide more information or give possible examples on the underlying physical processes

The inter-seasonal variation of the correlation coefficient linked to annual maximum water levels results from the marginal distribution of non-conditioned precipitation and surge. For example, in winter precipitation tends to be lower, so extreme water levels are mostly surge driven. Differently, in summer the likelihood of heavy precipitation increases, which increases the chance of compound surge and precipitation leading to extreme water levels. We will rephrase this information to make it clearer in the manuscript.

15. In line 297-298, you mention that separating the analysis in seasonal clusters did not lead to an improvement, but do not report to what extent. Please provide more information to the respective section. Additionally, in line 324 you mention that separating the statistical analysis in tidal clusters did not lead to an improvement. Please specify to what it did not lead to an improvement.

We separated the 800 annual events into seasons and into clusters (defined as a function of tidal range), respectively. None of these options led to a better performance of the impact function (in terms of RMSE, MAE and correlation coefficient) and return levels (visual inspection). We will add more details in the revised text, as suggested by the reviewer.

16. The section about seasonal variability evaluates the dependence structure of the predictors and reports the Kendall's rank correlation for the respective seasons. This is a very interesting read and discussion, however the authors report in line 298-299 that the spread of annual maxima events is uneven and that for some months few events occur. Have the authors considered restructuring the inland water levels maxima in seasonal maxima, resulting in 800 maxima inland water levels per season?

We considered performing a seasonal analysis but eventually discarded this option for two reasons. We wanted to replicate the results of van den Hurk et al (2015) where annual maxima is used. Also, using seasonal maxima might lead to consider non-extreme water level events, which are not the focus of this study.

17. Please provide contextualization on the results reported in lines 327-330.

These results are intended to give an overview of the effects of climate variability in the estimation of the correlation between predictors. We divided the dataset into subsets of 50 years and assessed the correlation for each subset. We found that correlation varies significantly among 50-year subsets and shortening the dataset can often lead to not sufficient data to get statistically significant correlation estimates. We will rephrase this paragraph to make it clearer.

Technical corrections: - In line 124-132, it would improve readability to also refer to the respective sections in the methods for the different steps of the conceptual model. - In line 200-202, extreme water levels exceeding 0 meter is used to describe the higher end of the water levels, however it would be more sensible for the reader at this stage to refer to in percentiles or flood warning level as indicated in the sup. - In line 260, do

you mean 'inland' water level? - Line 338 misses a word: 'in the following ...'. - Line 366 Fig S14 should be Fig S12 - Line 372 now reads as if empirical analysis consists of 100,000 events. Please rephrase. - In the caption of Fig 6, transparent should be added to 'Green illustrates the uncertainty ...' - In the caption of Figs 6 and 7 c, d, and e don't match up. - Table 4 subpanel e, to my understanding the copula of 800-year ensemble should be marked and not the total surge of 800-year ensemble. - Please clarify the sentence in line 414-415 starting from 'hence'. - In the supplementary information line 45 contains a duplicate of 'a'. - In the supplementary information lines 51-52 reports difference between shuffled dataset and dependent dataset while using the same symbols. As a suggestion the authors can use for example ðĺŘťðĺŘűmax,shuffled in order to increase readability. - The caption of Fig S7 contains a duplicate of 'that'.

We thank the reviewer for pointing out these technical corrections. We will address them in the manuscript appropriately.

---

## Author Comment (AC2) · 19 Jan 2021

We, the authors, thank the anonymous reviewer for providing a thorough and comprehensive review. We acknowledge their suggestions will help us improve the quality of the manuscript and hope the suggested amendments satisfy their concerns. Below we provide our point-by-point response to their comments.

1. In general, there are a lot of references to figures in the supplemental information that feel as if they are written in the same manner that one would normally refer to an in-text figure. If showing these figures are crucial to communicating the results, then I feel they should be in the main paper. Otherwise I suggest rewriting the sections (i.e.

[Figure]

4.1.2, 4.1.3, 4.1.4, etc.) to explain the results in words without referencing a take-away point that a reader would need to see a figure to understand. You can then tell the reader that further information is available in the supplement.

There are too many figures in the supplementary material to be included in the main text. Yet most of them provide relevant insights to the study. We will therefore rewrite Section 4 to better explain the results in words while keeping the references to the supplementary figures within brackets.

2. The introductory paragraph refers to the same author/lab groups efforts in 5 straight individual sentences. While subsequent paragraphs show the author's have a broad grasp on literature beyond this one lineage, I recommend broadening the background to highlight that the motivation for this work does not arise simply from one group's efforts. There are many other works that have identified and attempted to account for multivariate climate drivers of compounding events (e.g. Anderson et al. 2019, Serafin et al. 2014, Rueda et al. 2016).

The introduction provides a comprehensive list of studies from different groups, which is not meant to include all relevant compound analysis studies to date but to give an overview of the state of the art. However, we will broaden the background by including a few more references as suggested by the reviewer but without lengthening the introduction excessively.

3. I'll admit I am confused by the tidal variability included in Figure 1. The text at Line 114 indicates that the tide cycle is added but doesn't give any specifics (I suggest adding these specifics to improve transparency). Figure 1 makes it look like all 800 events had the maximum occur at the same phase of the tide? Otherwise the bold tidal level would be a more flat line with a large envelope of variability around it? If the events do all occur at the same tidal phase then that would be a significant limitation of this work. Perhaps Figure 1 is only a single example taken from the 800 annual and the text caption for the Figure could be rewritten to prevent the interpretation that it is

derived from all 800 scenarios.

As stated below in our answer to question 6, we will add more details about the data. An artificial extension of the historical astronomical tide between 1950 and 2000 was added to the modelled storm surge data. Figure 1 is a composite (average) plot and therefore not taken from a single event but derived from all 800 annual maximum water levels. The catchment around Lauwersmeer is a managed water system in which gates open during low tide allowing the water to discharge into the sea by gravity, as mentioned in lines 148-150. Therefore, it is reasonable that the annual maximum water level always occurs at approximately the same phase of the tide (close to the minimum tide). This is, however, not a limitation of this work. Most water managed systems are expected to have similar discharge strategies. On the other hand, the framework proposed is general and it can be applied to any water system, once the appropriate predictors have been identified. In the revised version of the manuscript we will add a paragraph to discuss the transferability of our modelling framework to other study areas

4. Are copulas fit to purely empirical distributions? At which point the underlying assumption is that the 800 events can accurately represent the tails of the distributions? If this is the assumption being made then I think it should be explicitly stated in the manuscript and acknowledged as a potential limitation for obtaining extremes.

Yes, copulas are fit to empirical distribution, so the choice of the copula does not depend on the marginal distribution. This implies that we assume that 800 events can accurately represent the correlation between large percentiles of the surge and precipitation predictors, but not the tails of the distributions of the surge and precipitation predictors, as we use marginal density functions to obtain the final joint probability density function. This is a common approach followed in previous studies, even when using significantly shorter records (e.g. Jane et al., 2020). In lines 315-322 of the manuscript we discussed the challenges encountered when assessing the degree of compoundness for large return periods (e.g., 800 years)

Reference: Jane, R., Cadavid, L., Obeysekera, J., & Wahl, T. (2020). Multivariate statistical modelling of the drivers of compound flood events in south Florida. Natural Hazards and Earth System Sciences, 20(10), 2681-2699.

5. Although a paragraph at the beginning of Section 2 does describe the study site, I think an annotated figure of the coast, the physical point where all data is obtained, and the square area or arial footprint of the watershed catching the precipitation could aid the manuscript. I was left wondering about the coastal configuration, proximity to open water, and proximity to human altered landscapes.

We agree with the Reviewer that a figure of the study site featuring the characteristics stated above may help readers to have a better understanding of the system. We will add a figure of the study area in the revised manuscript.

6. I think the paragraph between lines 112-120 could grow to be multiple paragraphs that detail the methodology from van den Hurk et al. (2015), as this manuscript is heavily dependent on that work.

We understand the suggestion of the Reviewer. However, we feel that adding multiple paragraphs talking about an already published work might seem redundant, as interested readers can refer to the other article. We will include more details about the methodology implemented by van den Hurk et al. (2015) while trying to be concise.

7. Although the explanation of copulas is suitable for publication, I think that the author's dynamical interpretation of the final copulas could be useful. By that I mean, why does a Frank copula fit better and what does that tell us about the dynamics of the compound hazard?

For the 2D case, we obtained a rotated Tawn copula (90 degrees) with negative (weak) correlation between the chosen predictors. As comprehensively explained in Section 4.1, the dynamical interpretation of the correlation coefficient is not straightforward. For example, a negative correlation between predictors does not lead to a negative

correlation between the underlying drivers (surge and precipitation) as the predictors are conditioned to the impact variable. Comparison with the shuffled data reveals that drivers are indeed positively correlated, although the correlation is not very strong and therefore does not lead to a positive correlation between the conditioned predictors. This, for example, agrees with obtaining a copula with asymptotic independence between the predictors. We will briefly comment on the main traits of the chosen copula in the revised version of the manuscript. However, we prefer not to extensively discuss the interpretation of the copula of conditioned predictors as it might contribute to confusion about the physical understanding of the underlying (unconditional) drivers.

8. I think the usefulness of this case study to readers may be improved by including a commentary on what physical processes are or are not being wrapped up into the relatively broad predictors. Does the original modeling framework exhibit sea level anomalies at longer frequencies than just meteorological surges and tides (i.e. monthly or seasonal anomalies)? Does the location of this virtual tide gauge experience waves? Perhaps a paragraph at the end of the discussion could address limitations and how extensible the study is to other sites.

We agree with the reviewer about clarifying framework transferability and limitations in our manuscript. Although the results presented here are site specific, the general framework can be transferred to other locations, given the availability of relatively long overlapping records of flooding drivers and impact variable. If the size of the database needs to be extended prior developing a multivariate statistical framework, a Regional Climate Model (RGM) and a hydrological management simulator to derive empirical estimates could be used (e.g., van den Hurk et al, 2015). Depending on the size and resolution of the RCM, appropriate computational resources may be required. Defining appropriate predictors that optimizes the performance of an impact function depends on the hydrological characteristics and management of a given system. For systems with low or no management, we would expect a more straightforward construction of an impact function, but appropriate lags between drivers and impacts should be accounted

for. Characterizing probability distributions that precisely describe the marginals and fitting copulas that accurately capture the dependence structure largely depend on data availability.

The proposed framework assumes waves are not an important driver of inland extreme water levels, and only low-frequency sea-level components are accounted for. This is reasonable considering the characteristics of the study area: 1) sheltering effects of barrier islands protecting from extreme wave climate and 2) shallow waters inducing wave breaking for large wave heights. On the contrary, surge is a relevant driver of extreme coastal water levels in such shallow water environments. However, if our framework were to be implemented in areas exposed to extreme waves, ocean wave predictors would need to be included in the model. Yet the proposed framework described in Section 3 would still be valid.

The surge is calculated from the meteorological forcing for all relevant time scales, from daily to multi-annual, using the empirical relationship between surge and model generated wind. Apart from the astronomical tide no other sources of variability are incorporated in the sea level records. Therefore, the main limitation of this study is the exclusion of long-term nonstationary sea level processes, such as sea-level rise which plays a large role in increasing extreme water levels (Taherkhani et al., 2020). However, since our focus is on the assessment of historical extreme sea level climate with focus on the effect of climate variability, this assumption is reasonable.

Reference: Taherkhani, M., Vitousek, S., Barnard, P. L., Frazer, N., Anderson, T. R., & Fletcher, C. H. (2020). Sea-level rise exponentially increases coastal flood frequency. Scientific reports, 10(1), 1-17.

---

## Author Response (AR1)

The manuscript 'Multivariate statistical modelling of extreme coastal water levels and the effect of climate variability: a case study in the Netherlands' assesses an impact function that can reproduce inland water levels in a human controlled system by event sampling and conditioning the drivers. By modelling the dependence structure between the different drivers to generate paired synthetic events, the authors are able to assess compounding effects of surge and precipitation on inland water levels. Overall, this study is an interesting read and I commend the authors for their nice work. It uses well-established methods and builds on previous assessments. Furthermore, it provides new insights in modelling compounding effects of surge and precipitation, and an interesting analysis of climate variability and using a short subset of the data. The manuscript also provides interesting and detailed information and discussion on underlying processes of the predictor selection and interpretation of the compounding effects of the two drivers. However, in its current form this study has a number of limitations that I would like to see addressed. For instance, the contextualization of using a case study in an area with a high degree of human management is lacking, steps undertaken in the methods section need more clarification, and decisions undertaken in the results need more clarification and transparency. Therefore, I propose to reconsider this manuscript for publication upon revision of the following issues.
Specific comments:

We, the authors, thank the anonymous reviewer for providing a thorough and comprehensive review. We acknowledge the suggestions have helped us improving the quality of the manuscript and hope our amendments satisfy his/her concerns. Below we provide our point-by-point response to his/her comments.

1. The title of this manuscript is framed as a case study that provides a statistical framework for assessing extreme coastal water levels and climate variability that can be used for other case studies as well. By framing the title like this I would expect a discussion in the manuscript that addresses how this statistical framework (e.g. conditioning the drivers) can be used for other areas of interest or even a different region in the Netherlands. This contextualization of how a user can use this framework in other areas of interest is lacking in the manuscript's current form.

In the revised version of the manuscript, we have clarified the transferability of the methodology. The statistical framework can be used to other areas of interest, given the availability of relatively long overlapping records of flooding drivers and impact variable. Defining appropriate impact-based predictors that optimize the performance of the impact functions depends on the hydrological characteristics and management of a given system. For systems with low to no management, we would expect a more straightforward construction of an impact function. In any case, composite (average) plots can guide this process where appropriate lags between drivers and impacts should be accounted for.

This study did not include wave-driven water levels (i.e. wave set up). This is a reasonable assumption in the shallow Wadden Sea (sheltered by barrier islands, see new Figure 1), where surge is the main flooding marine driver. In other locations, the wave contribution, or other drivers such as snow melt, might need to be considered as well. Additionally, we did not

account for low-frequency variations of water levels such as sea level rise, which would need to be considered if results where to be extrapolated into the future.

We have added a paragraph in the revised manuscript to discuss this transferability as well as limitations (see lines 537-558).

2. The case study in the Netherlands provides an analysis on an area with a high degree of human management. As the title of the manuscript does not cover this, I would suggest to either add this information to the title or add a short discussion on how this statistical framework can be used for other areas which do not have a high degree of human management.

We agree with the Reviewer suggestion and this information has been added to the new title and is also discussed in the main text (see lines 89-90, 209-213, 543-544). The new title is "Statistical modelling and climate variability of compound surge and precipitation events in a managed water system: a case study in the Netherlands".

3. Throughout the manuscript, water levels are most often referred to as inland water levels (line 1), however sometimes the authors use solely water levels without the adjective 'inland' (e.g. line 99), or extreme coastal water levels as is stated in the title of the manuscript. I would suggest staying consistent with the terminology and provide a clear description of the water levels (e.g. how much inland, coastal/inland water levels).

We agree with the reviewer about the ambiguity of our previously used terminology. We have modified the terminology throughout the paper in the following manner: Still water level, to refer to water elevation on the coastal side (not including waves); and inland water level, to refer to reservoir levels. Surge is still water level after subtracting tides. We hope it is clearer now.

4. While the manuscript discusses relevant previous studies in the introduction (line 59-73), the research gap is not pointed out clearly. As a consequence, the novel aspects of this study and the research gap do not come across strongly. Therefore, I suggest adding more detail to this section in the introduction.

The last paragraph of the Introduction mentioned the novel aspects of our study. However, to emphasize the novelty of our study in relation to existing gaps, we have expanded the last paragraph of the Introduction (lines 87-101), as suggested by the reviewer. We have also changed the title which now includes these novel aspects: 1) compound analysis in a managed water system, 2) sensitivity of such analysis to climate variability.

5. In order to improve readability, I would suggest to rephrase line 116 by using frequencies, i.e. more frequent in original data or less frequent in shuffled data.

We prefer to keep the term return period rather than frequencies, as it is a well-known term that is used in many contexts: risk analysis, impact assessment, infrastructure design, etc. However, we have rephrased the paragraph to improve readability (see lines 137-140).

6. In section 2, data and study area, please provide more background information how the predictors total surge and precipitation were derived. For instance, information how the surge and tide are added (van der Hurk et al., 2015).

The revised version of the manuscript provides now more information about how total surge (still water level in the revised manuscript) and precipitation were derived (see lines 126-133). In a nutshell, surge is calculated from wind speed via an empirical equation that was previously calibrated for the study area. The tidal time series is an artificial extension of the standard astronomical equations and was calculated using all known current tidal constituents for a complete period of 800 years. Total water level is the sum of surge and tide.

7. In line 137-138, you mention that the performance of the impact function is highly sensitive to the selection of the predictors, yet no sensitivity analysis or the degree of sensitivity is reported or shown. Please provide more information and details on how sensitive it is.

Lines 183-200 summarize the sets of predictors tested in this study. We have also added a few examples in the Supplementary Material to illustrate the sensitivity of the impact function performance to the selection of predictors (Fig S1), for example: Min Surge 12, 24, 36, 48, 60h, CumPrcp 5d, 12d; Max Surge 36h, Mean Surge 36h, etc.). The inland water level in this specific location is more sensitive to variations in storm surge than to variations of precipitation.

8. Please contextualize, if possible, why the annual maxima surge events are at least 3 days (5th percentile), see Figure 1b and lines 174-176.

It is reasonable to obtain from the composite analysis and impact function optimization that the relevant duration of storm surge is 3 days as this is the mean duration of cyclones over East-Central Europe (Bartoszek, 2017). Note that from Figure 1b (Figure 2b in the updated manuscript) we cannot conclude however that events linked to annual maxima surge events are at least 3 days long. The shaded area depicts values from $5^{th}$ to $95^{th}$ for each time considered. However, storm surge time series of individual events are not necessarily parallel to the mean storm surge, or lower/upper envelopes. As the lower envelope of storm surges is not necessarily linked to a single event, the associated duration below 3 days does not necessarily have a probability below 5 %. We have added a comment about the choice of 3 days for the storm surge predictor (see lines 204-205)

Reference:
Bartoszek, K. (2017) The main characteristics of atmospheric circulation over East-Central Europe from 1871 to 2010. Meteorology and Atmospheric Physics, 129, 113-129.

9. In lines 176-178, please provide more contextualization about the tradeoff and why the minimum total surge is selected.

Our case study is a water-managed system. To try to prevent inland water levels from reaching extreme levels, the system is regulated by opening the gates around low tide. However, if the still water level at the low tide is too high, gates cannot open, water cannot be released, and inland water level might increase due to precipitation and river runoff. Therefore, the annual maximum water level better relates to the previous local minimum of coastal water level (i.e. total storm surge), rather than the local maximum, as shown in the composite plot (Fig 2 in the updated manuscript). This differs from a natural water system in which the maximum or mean storm surge would probably be a better predictor to describe extreme water levels. In fact, when the tide and surge are separated, we found that the mean surge (and not the minimum surge) is a better predictor. The choice of 36 h is not randomly selected as a value in between 72h and 12h (from the 3D marine predictors) but is the result of optimizing the impact function performance, for which a wide range of time lags were tested. This choice is also supported by the visual composite analysis (Figure 2b).

We have rephrased the text to avoid confusion (see lines 201-213).

10. In Table1, the selected predictors of the two cases are reported, taking into account the three aspects mentioned in lines 141-145. Additionally, information is provided for the selection of predictors in lines 159-161. Please provide more information about the optimization technique used. Why was the maximum (next to the minimum and mean) for the conditioning not included? Which approach was used for this during conditioning of the drivers (MLR, MLRbin, ANN, etc.)? Is the performance of the predictor selections evaluated on the metrics used throughout this study, or the tradeoff between the metrics and visual inspection of the events that exceed the flood warning level as in line 222-223? What is the time step of additional hours prior to the event used for this selection? Were all possible combinations of the selected time steps and statistics evaluated or was an optimization technique used for this (e.g. random search)?

We have added more details of the performance assessment and optimization (lines 183-189, Figure S1). We tested a wide range of predictors, mean, max and min values, for different time lags. This selection was guided by the composite plots and physical understanding of the water system and tested by means of the impact function performance as well as the return level estimates.

11. In line 191-193, please provide short details on which architecture and hyper parameters are used for the machine learning approaches.

We have included the main information in the manuscript (lines 227-232).

In summary, the learning process of the artificial neural network used here is based on stochastic gradient descent, and the activation function is the sigmoid function. The architecture of the network is as follows: input layer with 2 (2D case) or 3 (3D case) neurons; 2 hidden layers with 8 neurons each, output layer with 1 neuron.

The number of trees in the random forest was set to 50, after performing a sensitivity analysis assessing the overall performance of the approach (estimated as root-mean-square error (RMSE) via k-fold validation approach) depending on the number of trees. We selected 50 because increasing the number of trees beyond that value did not lead to an increase in performance.

It is however important to notice that, we refrained from using additional (more sophisticated) machine learning approaches and testing other architectures because we achieved a reasonably good performance using regression for most return periods by means of bin-sampling. We believe this approach is easier to implement, which can aid transferability.

12. Like equation 2, are the predictors in equation 3 for the 3D case also standardized?

No, equation 3 for 3D case is not standardized. The standardization was only shown in Eq. 2 to preliminary demonstrate the importance of total surge (coastal water level in the revised manuscript) to drive extreme reservoir levels, as compared to precipitation. We added the standardized equation for the 3D case (Eq. 4 in line 261).

13. In lines 250-253, please provide more information on what terms the 3D case generally does not outperform the 2D case. To me it seems that the 3D case performs better on the reported metrics. Above the flood warning level, the differences looks only marginal (confidence interval of the 1000 bootstrap runs not reported). If the focus of this manuscript is on the distribution of extreme cases above the flood warning level, then it should be clearly stated in the manuscript. Additionally, lines 366-367 report that adding complexities does not necessarily improve performance. However, the reported metrics show an improvement. In lines 367-368 the authors report that the performance between the two cases differ slightly for higher return periods. Why did you choose to not report metrics (e.g. MAE) of those extremes of extreme events? Moreover, lines 440-441 report that the 3D case did not lead to an overall improvement. Pleas provide more information to the respective section why those decisions are taken and on what basis (e.g. define overall in overall improvement).

We have revised the text to better explain the comparison between 2D and 3D cases, and why we chose the 2D framework. This comparison is done in terms of the impact function performance (RMSE, MAE, etc), and return level estimates (visual inspection from the return plot, with or without considering the estimated joint probability density function). Please see lines 262-267, 427-432.

In summary, the 3D case performs better based on the reported metrics for the impact function, but the increase in performance compared to the 2D case is minimal (RMSE_3D = 0.085 m; RMSE_2D = 0.091 m). Also, 2D performs better for the larger return levels. When the joint density distribution is also accounted for, the 3D slightly outperforms the 2D (RMSE_3D = 0.019 m; RMSE_3D = 0.02 m) but we opt for the simpler case following the principal of parsimony. The residual improvement in the return levels does not justify an increase in model complexity. Additionally, 3D shows an increase of uncertainty when shorter datasets are used.

14. Line 291-292, please provide more information or give possible examples on the underlying physical processes

The inter-seasonal variation of the correlation coefficient linked to annual maximum water levels results from the marginal distribution of non-conditioned precipitation and surge, which links to the underlying physical processes mentioned in Section 2. For example, in winter (and specially in February and March) the likelihood of annual maximum accumulated precipitation drops (see Figure S7c), so extreme water levels are mostly surge driven. Differently, in summer the likelihood of heavy precipitation increases, which increases the chance of compound surge and precipitation leading to extreme water levels. We have rephrased this information to make it clearer in the manuscript. See lines 340-349.

15. In line 297-298, you mention that separating the analysis in seasonal clusters did not lead to an improvement, but do not report to what extent. Please provide more information to the respective section. Additionally, in line 324 you mention that separating the statistical analysis in tidal clusters did not lead to an improvement. Please specify to what it did not lead to an improvement.

We separated the 800 annual events into seasons and into clusters (defined as a function of tidal range), respectively. None of these options led to a better performance of the impact function (in terms of RMSE) and return levels (visual inspection and in terms of RMSE). We have added a few details in the revised text (see lines 350-355 and 379-380), as suggested by the reviewer.

16. The section about seasonal variability evaluates the dependence structure of the predictors and reports the Kendall's rank correlation for the respective seasons. This is a very interesting read and discussion, however the authors report in line 298-299 that the spread of annual maxima events is uneven and that for some months few events occur. Have the authors considered restructuring the inland water levels maxima in seasonal maxima, resulting in 800 maxima inland water levels per season?

We considered performing a seasonal analysis but eventually discarded this option for two reasons. We wanted to replicate the results of van den Hurk et al. (2015) where annual maxima

are used as sampling strategy. Also, using seasonal maxima might lead to consider non-extreme water level events, which are not the focus of this study. See lines 350-355.

17. Please provide contextualization on the results reported in lines 327-330.

These results are intended to give an overview of the effects of climate variability in the estimation of the correlation between predictors. We divided the dataset into subsets of 50 years and assessed the correlation for each subset.  We found that correlation varies significantly among 50-year subsets and shortening the dataset can often lead to not sufficient data to get statistically significant correlation estimates.

We have added more details to this paragraph (lines 382-388) to make this clearer, and we moved the related figure from SM to the main manuscript (now Figure 7).

Technical corrections:
- In line 124-132, it would improve readability to also refer to the respective sections in the methods for the different steps of the conceptual model.
We believe that since the remainder of Section 3 is dedicated to explain the steps of the methodology, there is no need to point to these subsections. However, we have rephrased the paragraph in lines 155-166 to make the process clearer.

- In line 200-202, extreme water levels exceeding 0 meter is used to describe the higher end of the water levels, however it would be more sensible for the reader at this stage to refer to in percentiles or flood warning level as indicated in the sup.
In the revised manuscript, we provide more details about the performance for levels above the warning level at the end of Section 3.3. (see lines 262-267).

- In line 260, do you mean 'inland' water level?
Yes. This sentence has been removed in the new version of the manuscript though.

- Line 338 misses a word: 'in the following …'.
Corrected.

- Line 366 Fig S14 should be Fig S12
This has been updated according to new numbering of SM figures.

- Line 372 now reads as if empirical analysis consists of 100,000 events. Please rephrase.
It has been rephrased.

- In the caption of Fig 6, transparent should be added to 'Green illustrates the uncertainty …'
This has been corrected.

- In the caption of Figs 6 and 7 c, d, and e don't match up.
This has been corrected.

- Table 4 subpanel e, to my understanding the copula of 800-year ensemble should be marked and not the total surge of 800-year ensemble.
This has been corrected.

- Please clarify the sentence in line 414-415 starting from 'hence'.
It has been clarified, please see lines 468-471.

- In the supplementary information line 45 contains a duplicate of 'a'.
This has been corrected.

- In the supplementary information lines 51-52 reports difference between shuffled dataset and dependent dataset while using the same symbols. As a suggestion the authors can use for example $AC_{max,shuffled}$ in order to increase readability.
The suggestion has been followed.

- The caption of Fig S7 contains a duplicate of 'that'.
This has been corrected.

The manuscript titled 'Multivariate statistical modelling of extreme coastal water levels and the effect of climate variability: a case study in the Netherlands' presents an interesting attempt to quantify the joint probability of coastal water levels and precipitation that ultimately create a compound flood hazard. They use 800 synthetic annual maxima events to define the marginals of copulas and create a trained impact function to relate predictands and predictors. The degree of uncertainty introduced by shorter records is also quantified as a commentary on the importance of data volume for such methodologies. While I do think the publication will ultimately be a quality contribution to the literature, it currently is vague on some methodology components that need further clarification. I recommend that the manuscript be returned to address the following thoughts:

We, the authors, thank the anonymous reviewer for providing a thorough and comprehensive review. We acknowledge the suggestions have helped us improving the quality of the manuscript and we hope our amendments satisfy his/her concerns. Below we provide our point-by-point response to his/her comments.

1. In general, there are a lot of references to figures in the supplemental information that feel as if they are written in the same manner that one would normally refer to an in-text figure. If showing these figures are crucial to communicating the results, then I feel they should be in the main paper. Otherwise I suggest rewriting the sections (i.e. 4.1.2, 4.1.3, 4.1.4, etc.) to explain the results in words without referencing a take-away point that a reader would need to see a figure to understand. You can then tell the reader that further information is available in the supplement.

There are too many figures in the supplementary material to be included in the main text. Yet, most of them provide relevant insights to the study. However, we understand the Reviewer suggestion, so to improve clarity while keeping the manuscript to a reasonable length, we have included figures S6 and S9 in the main manuscript, and have rewritten Section 4 to better explain the results in words while keeping the references to the remaining supplementary figures within brackets.

2. The introductory paragraph refers to the same author/lab groups efforts in 5 straight individual sentences. While subsequent paragraphs show the author's have a broad grasp on literature beyond this one lineage, I recommend broadening the background to highlight that the motivation for this work does not arise simply from one group's efforts. There are many other works that have identified and attempted to account for multivariate climate drivers of compounding events (e.g. Anderson et al. 2019, Serafin et al. 2014, Rueda et al. 2016).

We have edited the Introduction to address this concern by broadening the background (see lines 44-47). We hope the Introduction now better reflects the state of the art by including a diverse contribution from different groups.

3. I'll admit I am confused by the tidal variability included in Figure 1. The text at Line 114 indicates that the tide cycle is added but doesn't give any specifics (I suggest adding these specifics to improve transparency). Figure 1 makes it look like all 800 events had the maximum occur at the same phase of the tide? Otherwise the bold tidal level would be a more flat line with a large envelope of variability around it? If the events do all occur at the same tidal phase then that would be a significant limitation of this work. Perhaps Figure 1 is only a single example taken from the 800 annual and the text caption for the Figure could be rewritten to prevent the interpretation that it is derived from all 800 scenarios.

Following the Reviewer suggestion, we have added more details about the data (see also question 6). An artificial extension of the historical astronomical tide between 1950 and 2000 was added to the modelled storm surge data. Figure 1 (now Figure 2) is a composite (average) plot and therefore not taken from a single event but derived from all 800 annual maximum water levels. The catchment around Lauwersmeer is a managed water system in which gates open during low tide allowing the water to discharge into the sea by gravity, as it was mentioned in lines 148-150 of the old manuscript. Therefore, it is reasonable that the annual maximum water level always occurs at approximately the same phase of the tide (close to the minimum tide). This is, however, not a limitation of this work. Most water managed systems are expected to have similar discharge strategies. On the other hand, the framework proposed is general and it can be applied to any water system once the appropriate predictors have been identified.

In the revised version of the manuscript we have added a paragraph to discuss the transferability of our modelling framework to other study areas, as well as the limitations (see lines 537-558).

4. Are copulas fit to purely empirical distributions? At which point the underlying assumption is that the 800 events can accurately represent the tails of the distributions? If this is the assumption being made then I think it should be explicitly stated in the manuscript and acknowledged as a potential limitation for obtaining extremes.

Yes, copulas are fit to empirical distribution, so the choice of the copula does not depend on the marginal distribution. This implies that we assume that 800 events can accurately represent the correlation between large percentiles of the surge and precipitation predictors, but not the tails of the distributions of the surge and precipitation predictors, as we use marginal density functions to obtain the final joint probability density function. This is a common approach followed in previous studies, even when using significantly shorter records (e.g. Jane et al., 2020). Although this is a common assumption of studies of this type, we have acknowledged it

in lines 280-285 in the revised manuscript. In lines 315-322 of the old manuscript (now lines 370-377) we discussed the challenges encountered when assessing the degree of compoundness for large return periods (e.g., 800 years).

Reference:
Jane, R., Cadavid, L., Obeysekera, J., & Wahl, T. (2020). Multivariate statistical modelling of the drivers of compound flood events in south Florida. Natural Hazards and Earth System Sciences, 20(10), 2681-2699.

5. Although a paragraph at the beginning of Section 2 does describe the study site, I think an annotated figure of the coast, the physical point where all data is obtained, and the square area or arial footprint of the watershed catching the precipitation could aid the manuscript. I was left wondering about the coastal configuration, proximity to open water, and proximity to human altered landscapes.

We agree with the reviewer that a figure of the study site featuring the characteristics stated above may help readers to have a better understanding of the system. We have added a figure of the study area in the revised manuscript, following the reviewer suggestions (now Fig 1).

6. I think the paragraph between lines 112-120 could grow to be multiple paragraphs that detail the methodology from van den Hurk et al. (2015), as this manuscript is heavily dependent on that work.

We understand the suggestion of the reviewer. However, we feel that adding multiple paragraphs talking about an already published work might seem redundant, as interested readers can refer to the other article. We have included more details about the methodology implemented by van den Hurk et al. (2015) while trying to be concise (see lines 126-133).

7. Although the explanation of copulas is suitable for publication, I think that the author's dynamical interpretation of the final copulas could be useful. By that I mean, why does a Frank copula fit better and what does that tell us about the dynamics of the compound hazard?

For the 2D case, we obtained a rotated Tawn copula (90 degrees) with negative correlation between the chosen predictors. As comprehensively explained in Section 4.1, the dynamical interpretation of the correlation coefficient is not straightforward. For example, a negative correlation between predictors does not lead to a negative correlation between the underlying drivers (surge and precipitation) as the predictors are conditioned to the impact variable. Comparison with the shuffled data reveals that drivers are indeed positively correlated, although the correlation is not very strong and therefore does not lead to a positive correlation

between the conditioned predictors. We have briefly commented on the main traits of the chosen copula in the revised version of the manuscript (see Section 4.2.1).

8. I think the usefulness of this case study to readers may be improved by including a commentary on what physical processes are or are not being wrapped up into the relatively broad predictors. Does the original modeling framework exhibit sea level anomalies at longer frequencies than just meteorological surges and tides (i.e. monthly or seasonal anomalies)? Does the location of this virtual tide gauge experience waves? Perhaps a paragraph at the end of the discussion could address limitations and how extensible the study is to other sites.

We agree with the reviewer about clarifying framework transferability and limitations in our manuscript (see lines 537-558).

Lines 537-558: *Although the results presented here are site specific, the general framework can be transferred to other locations, given the availability of relatively long overlapping records of flooding drivers and impact variable. If the size of the database needs to be extended prior to developing a multivariate statistical framework, a regional climate model (RCM) SMILE and a hydrological management simulator to derive empirical estimates could be used (e.g., van den Hurk et al., 2015). Depending on the size of the ensemble and spatial resolution of the RCM, large computational resources may be required. Defining appropriate predictors leading to a satisfying performance of the impact function depends on the hydrological characteristics and management of a given system. For systems with low or no management, we would expect a more straightforward construction of an impact function, but appropriate lags between drivers and impacts should be accounted for. Characterizing probability distributions that precisely describe the marginals and fitting copulas that accurately capture the dependence structure largely depend on data availability.*

*The proposed framework assumes waves are not an important driver of extreme IWLs, and only low-frequency sea-level components are accounted for. This is reasonable considering the characteristics of the study area: 1) sheltering effects of barrier islands protecting from extreme wave climate and 2) shallow waters inducing wave breaking for large wave heights. In contrast, surge is a relevant driver of extreme SWLs in such shallow water environments. However, if our framework were to be implemented in areas exposed to extreme waves, ocean wave predictors would need to be included in the model. Yet the proposed framework described in Section 3 would still be valid.*

*The surge is calculated from the meteorological forcing for all relevant time scales, from daily to multi-annual, using the empirical relationship between surge and model generated wind. Apart from the astronomical tide, no other sources of variability are incorporated in the sea level*

*records. Therefore, the main limitation of this study is the exclusion of long-term nonstationary sea-level processes, such as sea-level rise which plays a large role in increasing extreme SWLs (Taherkhani et al., 2020b). However, since our focus is on the assessment of historical extreme sea-level climate with focus on the effect of climate variability, this assumption is reasonable.*

Reference:

Taherkhani, M., Vitousek, S., Barnard, P. L., Frazer, N., Anderson, T. R., & Fletcher, C. H. (2020). Sea-level rise exponentially increases coastal flood frequency. Scientific reports, 10(1), 1-17.